# Efficacy and safety of metabolic interventions for the treatment of severe COVID-19: in vitro, observational, and non-randomized open-label interventional study

**Avner Ehrlich[1,2], Konstantinos Ioannidis[1,2], Makram Nasar[3], Ismaeel Abu Alkian[3], Yuval Daskal[1,2], Nofar Atari[4], Limor Kliker[4], Nir Rainy[5], Matan Hofree[6], Sigal Shafran Tikva[5,7,8], Inbal Houri[9], Arrigo Cicero[10], Chiara Pavanello[11,12], Cesare R Sirtori[12], Jordana B Cohen[13], Julio A Chirinos[13], Lisa Deutsch[14], Merav Cohen[1,2], Amichai Gottlieb[3], Adina Bar-Chaim[5], Oren Shibolet[15], Michal Mandelboim[15], Shlomo L Maayan[3], Yaakov Nahmias[1,2]\***

[1]Grass Center for Bioengineering, Benin School of Computer Science and Engineering, Jerusalem, Israel; [2]Department of Cell and Developmental Biology, Silberman Institute of Life Sciences, Jerusalem, Israel; [3]Division of Infectious Diseases, Barzilai Medical Center, Ashkelon, Israel; [4]Central Virology Laboratory, Public Health Services, Ministry of Health and Sheba Medical Center, Tel Hashomer, Israel; [5]Laboratory Division, Shamir (Assaf Harofeh) Medical Center, Zerifin, Italy; [6]Klarman Cell Observatory, The Broad Institute of Harvard and MIT, Cambridge, United States; [7]Hadassah Research and Innovation Center, Jerusalem, Israel; [8]Department of Nursing, Faculty of School of Life and Health Sciences, The Jerusalem College of Technology Lev Academic Center, Jerusalem, Israel; [9]Department of Gastroenterology, Sourasky Medical Center, Tel Aviv, Israel; [10]IRCSS S.Orsola-Malpighi University Hospital, Bologna, Italy; [11]Centro Grossi Paoletti, Dipartimento di Scienze Farmacologiche e Biomolecolari, Università degli Studi di Milano, Milano, Italy; [12]Centro Dislipidemie, Niguarda Hospital, Milano, Italy; [13]Perelman School of Medicine, University of Pennsylvania, Philadelphia, United States; [14]BioStats Statistical Consulting Ltd, Modiin, Israel; [15]Sackler Faculty of Medicine, Tel Aviv University, Tel Aviv, Israel

**\*For correspondence:**
ynahmias@gmail.com

## Abstract

**Background:** Viral infection is associated with a significant rewire of the host metabolic pathways, presenting attractive metabolic targets for intervention.

**Methods:** We chart the metabolic response of lung epithelial cells to SARS-CoV-2 infection in primary cultures and COVID-19 patient samples and perform in vitro metabolism-focused drug screen on primary lung epithelial cells infected with different strains of the virus. We perform observational analysis of Israeli patients hospitalized due to COVID-19 and comparative epidemiological analysis from cohorts in Italy and the Veteran's Health Administration in the United States. In addition, we perform a prospective non-randomized interventional open-label study in which 15 patients hospitalized with severe COVID-19 were given 145 mg/day of nanocrystallized fenofibrate added to the standard of care.

**Results:** SARS-CoV-2 infection produced transcriptional changes associated with increased glycolysis and lipid accumulation. Metabolism-focused drug screen showed that fenofibrate reversed lipid accumulation and blocked SARS-CoV-2 replication through a PPARα-dependent mechanism in both alpha and delta variants. Analysis of 3233 Israeli patients hospitalized due to COVID-19 supported in vitro findings. Patients taking fibrates showed significantly lower markers of immunoinflammation and faster recovery. Additional corroboration was received by comparative epidemiological analysis from cohorts in Europe and the United States. A subsequent prospective non-randomized interventional open-label study was carried out on 15 patients hospitalized with severe COVID-19. The patients were treated with 145 mg/day of nanocrystallized fenofibrate in addition to standard-of-care. Patients receiving fenofibrate demonstrated a rapid reduction in inflammation and a significantly faster recovery compared to patients admitted during the same period.

**Conclusions:** Taken together, our data suggest that pharmacological modulation of PPARα should be strongly considered as a potential therapeutic approach for SARS-CoV-2 infection and emphasizes the need to complete the study of fenofibrate in large randomized controlled clinical trials.

**Funding:** Funding was provided by European Research Council Consolidator Grants OCLD (project no. 681870) and generous gifts from the Nikoh Foundation and the Sam and Rina Frankel Foundation (YN). The interventional study was supported by Abbott (project FENOC0003).

**Clinical trial number:** NCT04661930.

## Editor's evaluation

In this study, a metabolism-related drug screen showed that fenofibrate reversed lipid accumulation and blocked SARS-CoV-2 replication through a PPARα-dependent mechanism in both α and δ variants. Patients taking fibrates displayed significantly lower markers of inflammation and experienced faster recovery from disease. The data offer significant support for the concept that PPARα should be considered as a potential therapeutic approach for SARS-CoV-2 infection and emphasizes the need to complete studies of fenofibrate in large randomized controlled clinical trials.

## Introduction

The severe acute respiratory syndrome coronavirus 2 (SARS-CoV-2) is a positive-strand RNA virus of the *sarbecovirus subgenus* that is related to SARS. SARS-CoV-2 infection leads to the development of coronavirus disease (COVID-19), an inflammatory lung condition resulting in acute respiratory distress and organ failure (*Grasselli et al., 2020*). SARS-CoV-2 has infected over 265 million individuals worldwide, causing nearly 5.3 million deaths since its emergence. Like other viruses, SARS-CoV-2 must rely on the host machinery to propagate, rewiring cellular metabolism to generate macromolecules needed for virion replication, assembly, and egress.

Recent work suggests that COVID-19 progression is dependent on metabolic mechanisms. Elevated blood glucose, obesity, and hyperlipidemia were found to be risk factors for SARS-CoV-2-induced acute respiratory distress, independently from diabetes (*Bornstein et al., 2020*; *Zhu et al., 2020b*). In fact, metabolic risk factors are associated with a more than 3-fold increase in COVID-19 severity risk, whereas inflammatory lung diseases, such as chronic obstructive pulmonary disease (COPD), and asthma are associated with less than a 1.5-fold increase in risk (*Ko et al., 2020*; *Williamson et al., 2020*).

Metabolomics of COVID-19 patient sera showed alterations in circulating amino acids, glucose, and lipids, correlated with changes in inflammation and renal function (*Thomas et al., 2020*). Work on SARS-CoV-2 infected monocytes showed raised glycolysis (*Codo et al., 2020*; *Ajaz et al., 2021*), whereas proteomics of infected kidney and colon cells showed that SARS-CoV-2 proteins interact with mitochondria, glycolysis, and lipid metabolism (*Gordon et al., 2020*; *Bojkova et al., 2020*). Other transcriptional analyses showed SARS-CoV-2 induced significant changes in similar metabolic pathways (*Delorey et al., 2021*; *Melms et al., 2021*; *Islam and Khan, 2020*; *Singh et al., 2021*). These results support earlier observations that the closely related SARS and MERS infections are reliant on altered lipid metabolism (*Yuan et al., 2019*; *McBride and Machamer, 2010*; *Yan et al., 2019*). However, recent clinical studies show conflicting results regarding the role of triglycerides in COVID-19 progression (*Barberis et al., 2020*; *Wang et al., 2020*; *Masana*

*et al., 2021*). While these data suggest that lipid metabolic interventions should be studied in the context of COVID-19, the current reliance on animal experiments limits such efforts due to critical differences in lipid metabolism between humans and rodents (*Bergen and Mersmann, 2005*; *Demetrius, 2005*).

Alarmingly, evidence from previous coronavirus outbreaks suggests that the metabolic rewiring induced by infection has detrimental and long-term effects post-recovery. MERS infection was associated with long-term immune dysregulation and enhanced susceptibility to metabolic diseases (*Kulcsar et al., 2019*), while SARS infection was associated with long-term alterations in lipid metabolism, hyperlipidemia, and hyperglycemia even 12 years post-recovery (*Wu et al., 2017*; *Yang et al., 2010*). Recent work points to similar post-sequelae effects of COVID-19 (*Akter et al., 2020*; *Al-Aly et al., 2021*; *Li et al., 2021*).

In this report, we charted the metabolic response of primary lung bronchiole and small airway epithelial cells to SARS-CoV-2 infection validating our results with multiple COVID-19 patient samples. We demonstrate intracellular lipid accumulation driven in part by the inhibition of PPARα-dependent lipid catabolism. Screening pharmacological modulators of the SARS-CoV-2 metabolic landscape showed that fenofibrate, and other PPARα-agonists that induce lipid catabolism, reversed metabolic changes and blocked SARS-CoV-2 replication in vitro. An observational study in 3,233 Israeli patients hospitalized due to COVID-19 was consistent with the in vitro observations, showing lower inflammation and faster recovery in patients taking fibrates, while those taking thiazolidinediones that lead to increased lipid accumulation in certain tissues (*Ahmadian et al., 2013*; *Todd et al., 2007*; *Phan et al., 2017*) exhibited worse outcomes. Additional validation was received by comparative epidemiological analysis from cohorts in Italy and the Veteran's Health Administration in the United States.

Moreover, we performed a prospective non-randomized interventional open-label study in which 15 patients hospitalized with severe COVID-19 were given 145 mg/day of nanocrystallized fenofibrate added to the standard of care. These patients demonstrated a rapid reduction in inflammation and a significantly faster recovery compared to patients admitted during the same period and treated with the same standard-of-care. This work demonstrates that pharmacological modulations of PPARα may be an effective treatment for coronavirus infection. The clinical translation of these findings can only be determined following randomized placebo-controlled clinical studies, which are currently ongoing in several international centers.

## Methods
### Experimental model and subject details
#### Human subjects
All protocols involving human tissue were reviewed and exempted by The Hebrew University of Jerusalem, the Israeli Ministry of Health, Sheba Medical Center and Icahn School of Medicine at Mount Sinai Institutional Review Boards.

Experiments using samples from human subjects were conducted in accordance with local regulations and with the approval of the institutional review board at the Icahn School of Medicine at Mount Sinai under protocol HS#12–00145 and the institutional review board at Sheba Medical Center under protocol SMC-7875–20.

All procedures performed in studies involving human participants were in accordance with the ethical standards of the institutional and/or national research committee and with the 1964 Helsinki Declaration and its later amendments or comparable ethical standards.

In the observational studies - the Israeli study was approved by the local institutional review board of the Hadassah Medical Center (IRB approval number no. HMO 0247–20) and the local institutional review board of the Ichilov Medical Center (IRB approval number no. 0282–20-TLV). The Italian study was reviewed by the local ethical board (AVEC) of the IRCSS S.Orsola-Malpighi University Hospital (approval number LLD-RP2018).

The interventional study was conducted in accordance with the Good Clinical Practice guidelines of the International Council for Harmonisation E6 and the principles of the Declaration of Helsinki or local regulations, whichever afforded greater patient protection. The study was reviewed and approved by the Barzilai Medical Center Research Ethics Committee (0105–20-BRZ).

## Cell culture

Normal human bronchial epithelial (NHBE) cells (Lonza, CC-2540 Lot# 580580), isolated from a 79-year-old Caucasian female and were maintained at 37 °C and 5% $CO_2$ in bronchial epithelial growth medium (Lonza, CC-3171) supplemented with SingleQuots (Lonza, CC-4175) as per manufacturer's instructions. Cells were maintained at the BSL3 facilities of the Icahn School of Medicine at Mount Sinai. NHBE cells (ATCC, PCS-300–010 Lot#63979089; #70002486), isolated from a 69-year-old Caucasian male and a 14-year-old Hispanic male were maintained in airway epithelial cell basal medium (ATCC, PCS-300–030) supplemented with Bronchial Epithelial Growth Kit as per the manufacturer's instructions (ATCC, PCS-300–040) at 37 °C and 5% $CO_2$. Cells were maintained at the BSL2 facilities of The Hebrew University of Jerusalem and the BSL3 facility of the central virology laboratory of the ministry of health and Sheba Medical Center.

Cells were authenticated at the source and routinely screened for mycoplasma using PCR.

## Viruses

SARS-related coronavirus 2 (SARS-CoV-2), Isolate USA-WA1/2020 (NR-52281) was deposited by the Center for Disease Control and Prevention and obtained through BEI Resources, NIAID, NIH. SARS-CoV-2 was propagated in Vero E6 cells in DMEM supplemented with 2% Fetal Bovine Serum (FBS), 4.5 g/L D-glucose, 4 mM L-glutamine, 10 mM Non-Essential Amino Acids (NEAA), 1 mM Sodium Pyruvate, and 10 mM HEPES. Infectious titers of SARS-CoV-2 were determined by plaque assay in Vero E6 cells in Minimum Essential medium (MEM) supplemented with 4 mM L-glutamine, 0.2% Bovine Serum Albumin (BSA), 10 mM HEPES and 0.12% $NaHCO_3$, and 0.7% agar.

Isolate hCoV-19/Israel/CVL-45526-NGS/2020 (alpha) and hCoV-19/Israel/CVL-12806/2021 (delta) were isolated from nasopharyngeal samples of SARS-CoV-2 positive individuals which contained the alpha sub-lineage B.1.1.50 (hCoV-19/Israel/CVL-45526-NGS/2020) and Delta B.1.617.2 (hCoV-19/Israel/CVL-12804/2021) variants by the central virology laboratory of the ministry of health and Sheba Medical Center. Confluent Vero E6 cells were incubated for one hour at 33 °C with the nasopharyngeal samples, followed by the addition of MEM-EAGLE supplemented with 2% Fetal Bovine Serum (FBS). Upon cytopathic effect detection, supernatants were aliquoted and stored at –80 °C. Infectious titers of SARS-CoV-2 were determined by a 50% endpoint titer (TCID50) for each variant in Vero E6 cells. Approximately $1×10^5$ Vero E6 cells were seeded and incubated at 37 °C for 24 hr. At that point, the cells were infected by 10-fold serial dilutions of each variant in MEM-EAGLE supplemented with 2% Fetal Bovine Serum (FBS). A Gentian Violet staining was used to determine the TCID50 of each variant, calculated using the Spearman-Karber method.

All work involving live SARS-CoV-2 was performed in the CDC/USDA-approved BSL3 facility of the Global Health and Emerging Pathogens Institute at the Icahn School of Medicine at Mount Sinai or in the BSL3 facility of the central virology laboratory of the ministry of health and Sheba Medical Center in accordance with institutional and national biosafety requirements.

## Methods details

### Analysis of gene expression by RNAseq

Expression count matrices were retrieved from GEO: GSE147507-Series1 (Bronchial; culture), GSE153970 (Small airway; culture), GSE147507-Series15 (Autopsy), GSE145926- (Lavage). Differential gene expression analysis was performed using a Poisson-Tweedie distribution model using the tweeDEseq Bioconductor package (*Esnaola et al., 2013*). Count data from GEO were normalized using a trimmed-mean of M values (TMM) normalization with the edgeR Bioconductor packages (*Robinson et al., 2010*). Data from GSE153970 was previously normalized in GEO and was not further normalized. Genes with the following criteria were considered differentially expressed: (1) p-value adjusted by B&H method FDR <0.05, (2) A fold change >1.25, (3) Minimal mean expression >20 in either condition (*Supplementary file 1*).

Bronchial culture samples are 3 independent primary normal human bronchial epithelial cultures infected apically with SARS-CoV-2 (USA-WA1/2020; MOI 2) for 24 hr, compared with three independent primary normal human bronchial epithelial Mock-infected with PBS for 24 hr.

Small airway culture samples are three independent primary human airway epithelial cultures infected apically with SARS-CoV-2 (MOI 0.25) for 48 hr, compared with three independent primary human airway epithelial cultures Mock-infected with PBS for 48 hr.

The autopsy samples are of two old (age >60) unidentified COVID-19 human subjects, who died due to COVID-19, had autopsy biopsy tissue acquisition post-mortem in Weill Cornell Medicine, and were provided as fixed samples for RNA extraction; the samples were compared with two old (age >60) unidentified human biopsy lung samples, taken during lung surgery and stored at Mount Sinai Institutional Biorepository and Molecular Pathology Shared Resource Facility (SRF) in the Department of Pathology, similarly provided as fixed samples for RNA extraction.

COVID-19 patients' lung epithelial cells are bronchoalveolar lavage fluid isolates from one severe case and five critical cases. The median age of the patients was 62.5 years, and the participants included four male and two female patients. All patients had Wuhan exposure history and had a cough and/or fever as the first symptom. Diagnosis of SARS-CoV-2 was based on clinical symptoms, exposure history, chest radiography and SARS-CoV-2 RNA-positive using commercial quantitative PCR with reverse transcription (qRT–PCR) assays. The samples were compared to three healthy donor controls. The median age was 24 years, and the participants included one female and three male patients. These donors were confirmed to be free of tuberculosis, tumor, and other lung diseases through CT imaging and other laboratory tests.

### Analysis of canonical splice variants

Reads were downloaded from SRA (GSE147507), and filtered and trimmed to remove low-quality reads and sequencing artifacts with fastp v20 (*Chen et al., 2018*) (https://github.com/OpenGene/fastp.git; *Chen, 2022*). Reads were pseudoaligned to the GRCh38 genecode human transcriptome (GRCh38.p13, version 32) using Kallisto version 0.46.1 (*Bray et al., 2016*; https://github.com/pachterlab/kallisto; *Sullivan, 2022*) run with the default k-mer length of 31, in single-read, single-overhang mode, with fragment mean length of 400 and 100 SD. Differentially expressed transcripts/genes were identified using Sleuth based on a likelihood ratio test comparing the condition of interest and 100 Kallisto bootstrap samples.

### Assembly of metabolic categories

Aggregate metabolic categories were created as previously described (*Levy et al., 2016*). Briefly, functional annotation gene-sets, taken from GO and KEGG, were merged into a set of glucose, lipid, mitochondrial, and amino acid gene-sets.

### Processing, analysis, and graphic display of genomic data

Hierarchical clustering, heat maps, correlation plots, and similarity matrices were created in Morpheus. Gene ontology enrichment analyses and clustering were performed using DAVID Informatics Resources 6.7 (*Huang et al., 2009*) and PANTHER Classification System (*Mi et al., 2019*). Metabolic network maps were created using McGill's Network Analyst Tool using the KEGG database (*Xia et al., 2015*).

### Quantification of intracellular glucose

To detect glucose uptake, we used 2-(N-(7-Nitrobenz-2-oxa-1,3-diazol-4-yl) Amino)–2-Deoxyglucose (2-NDBG) a fluorescent analog of glucose (Invitrogen, USA; N13195). 2-NDBG is transported through SGLT-1 and GLUT-2. Increased uptake leads to 2-NDBG accumulation in the cells. Cells infected with SARS-CoV-2 for 96 hr were exposed to 6 mM of 2-NDBG for 24 hr. Cells were then fixed, counterstained with 1 µg/mL Hoechst 33258. Staining intensity was normalized to Hoechst 33258 across multiple fields of view.

### Quantification of lipids

Lipid accumulation was measured using HCS LipidTOX Phospholipidosis and Steatosis Detection Kit according to the manufacturer's instructions (ThermoFisher, USA; H34158). Briefly, cells were incubated in complete bronchial epithelial growth medium supplemented with 1 x phospholipidosis detection reagent for 48 hr. Cells were subsequently fixed in 4% PFA and stained with 1 X neutral lipid detected reagent for 30 min and counterstained with 1 µg mL-1 Hoechst 33258. Staining intensity was normalized to the amount of Hoechst 33258 positive nuclei across multiple fields of view.

## Metabolic analysis of glucose, lactate, and glutamine

Metabolic analysis of SARS-CoV-2 infected culture medium in the BSL3 facility was done using Accu-trend Plus multiparameter meter (Roche Diagnostics). Culture medium was collected every 48 hr and stored at –80 °C prior to analysis. Measurements were carried out using Accutrend Plus Glucose and BM-Lactate Test Strips according to the manufacturer's instructions. Each measurement was done in 3 technical measurements for each sample, validated throughout the process using calibration medium. Glucose uptake, as well as lactate production, were calculated based on the difference between sample and control medium.

Metabolic analysis of SARS-CoV-2 proteins expressing culture medium in the BSL2 facility was done using amperometric glucose, lactate, and glutamine sensor array (IST, Switzerland) as previously described (*Ehrlich et al., 2018*). Each measurement was done in three technical measurements for each sample, calibrated periodically throughout the process using calibration medium, according to the manufacturer's recommendations. Glucose and Glutamine uptake, as well as lactate production, were calculated based on the difference between sample and control medium.

## Generation lentiviral SARS-CoV-2 constructs

Plasmids encoding the SARS-CoV-2 open-reading frames (ORFs) and eGFP control are a kind gift of Nevan Krogan (Addgene plasmid #141367–141395). Plasmids were acquired as bacterial LB-agar stabs and used per the provider's instructions. Briefly, each stab was first seeded into agar LB (Bacto Agar; BD, USA) in 10 cm plates. Then, single colonies were inoculated into flasks containing LB (BD Difco LB Broth, Lennox; BD, USA) and 100 µg/ml penicillin (BI, Israel). Transfection-grade plasmid DNA was isolated from each flask using the ZymoPURE II Plasmid Maxiprep Kit (Zymo Research, USA) according to the manufacturer's instructions.

HEK 293T cells (ATCC, USA) were seeded in 10 cm cell culture plates at a density of $4 \times 10^6$ cells/plate. The cells were maintained in 293T medium composed of DMEM high glucose (4.5 g/l; Merck, USA) supplemented with 10% FBS (BI, Israel), 1 x NEAA (BI, Israel), and 2 mM L-alanine-L-glutamine (BI, Israel).

The following day, cells were transfected with a SARS CoV 2 orf-expressing plasmid and the packaging plasmids using the TransIT-LT1 transfection reagent (Mirus Bio, USA) according to the provider's instructions. Briefly, 6.65 µg SARS CoV 2 lentivector plasmid, 3.3 µg pVSV-G, and 5 µg psPAX2 were mixed in Opti-MEM reduced serum medium (Gibco, USA), with 45 µl of TransIT-LT1, kept at room temperature to complex and then added to each plate. Following 18 hr of incubation, the transfection medium was replaced with 293T medium and virus-rich supernatant was harvested after 48 hr and 96 hr. The supernatant was clarified by centrifugation (500×g, 5 min) and filtration (0.45 µm, Millex-HV, MerckMillipore). All virus stocks were aliquoted and stored at –80 °C.

The packaging plasmids (psPAX2 and pVSV-G) are a kind gift from Prof. N. Benvenisti, Stem Cell Unit at The Hebrew University, Jerusalem, Israel.

## SARS-CoV-2 proteins lentiviral transduction

Approximately $1 \times 10^5$ cells were infected in two consecutive sessions of 12 hr each. A 50% dilution of the viral stock was used in both for a final transduction efficiency of about 60%. Transduction efficiency was validated by microscopy of the eGFP transduced culture.

## Metabolic flux quantification (Seahorse)

Mitochondrial Stress Test (Agilent; 103010–100) assay was conducted per manufacturer instructions as previously described (*Levy et al., 2016*). Briefly, cells were incubated in unbuffered DMEM supplemented with 2 mM glutamine, 1 mM sodium pyruvate, and 10 mM glucose (pH 7.4) for 1 hr at 37 °C in a non-$CO_2$ incubator. Basal oxygen consumption rate (OCR) was measured for 30 min, followed by injection of 1.5 µM oligomycin, a mitochondrial Complex V inhibitor that blocks oxidative phosphorylation. The decrease in OCR due to oligomycin treatment is defined as the oxidative phosphorylation rate. 0.5 µM carbonyl cyanide-4 (trifluoromethoxy) phenylhydrazone (FCCP), an uncoupling agent, is added at 60 min to measure maximal mitochondrial activity followed by complete inhibition at 90 min using a mixture of 0.5 µM antimycin A and rotenone, mitochondrial Complex III and Complex I inhibitors.

Free fatty acid oxidation was measured using XF Long Chain Fatty Acid Oxidation Stress Test Kit (Agilent; 103672–100) as previously described (**Levy et al., 2016**). Briefly, cells were incubated overnight in a substrate-limited medium containing 0.5 mM glucose, 1 mM glutamine, and 0.5 mM L-Carnitine to prime cells for exogenous fatty acid utilization. Basal OCR was measured in the presence of BSA-palmitate (C16:0) or BSA-control for 30 min, followed by sequential exposure to 4 μM etomoxir, a carnitine palmitoyltransferase I (CPT1) inhibitor, or medium, 1.5 μM oligomycin, 0.5 μM carbonyl cyanide-4 (trifluoromethoxy) phenylhydrazone (FCCP), and a mixture of 0.5 μM antimycin A and rotenone at 30-min intervals. Free fatty acid oxidation capacity was defined as the difference between spare capacity by etomoxir-treated and untreated conditions.

## Generation PPARα CRISPR knock-out cells

The PPARα knock-out cells were created using a Cas9-based, CRISPR system. Two different sgRNA oligos from the human GeCKO v.2 Human CRISPR Knockout Pooled Library (Addgene; #1000000048), PPARa HGLibA_37838 and HGLibB_37787, were cloned into the lentiCRISPR v2 plasmid (Addgene; #52961). The sgRNA cloning was performed according to the human GeCKO v.2 system instructions as previously described (**Liu et al., 2020**). Briefly, two oligos comprising each sgRNA insert were synthesized with BsmBI-compatible ends, and the vector plasmid was digested with BsmBI (FastDigest Esp3I, FD0454, Thermo), de-phosphorylated (FastAP thermosensitive alkaline phosphatase, EF0651, Thermo), and gel extracted (QiaQuick gel extraction, Qiagen). The sgRNA oligos were phosphorylated and annealed in a single session: first phosphorylation using T4 PNK (NEB-M0201S) followed by heating to 95 °C for 5 min and controlled cooling to allow annealing. The vector and insert fragments were ligated (T4 DNA ligase, EL0011) and transformed into chemically competent Stbl3 cells (Mix & Go! *E. coli* Transformation Kit, T3001, Zymo). Correctly ligated plasmids were used for lentiviral sgRNA vector production, as described before (**Liu et al., 2020**). Approximately $1×10^6$ cells were infected in two consecutive sessions of 12 hr each. The cells were then selected using 3 μM puromycin for 72 hr (Merck; P9620).

The lentiCRISPR v2 plasmid is a kind gift from Prof. N. Benvenisti, Stem Cell Unit at the Hebrew University, Jerusalem, Israel.

## RNA-Seq of viral infections

Approximately $1×10^5$ NHBE cells were infected with SARS-CoV-2 at a MOI of 2 (USA-WA1/2020) or TCID100 (hCoV-19/Israel/CVL-45526-NGS/2020 and hCoV-19/Israel/CVL-12806/2021) for 24 hr in complete bronchial epithelial growth medium. Total RNA from infected and mock-infected cells was extracted using TRIzol Reagent (Invitrogen) and Direct-zol RNA Miniprep kit (Zymo Research) according to the manufacturer's instructions and treated with DNase I. RNA-seq libraries of polyadenylated RNA were prepared using the TruSeq RNA Library Prep Kit v2 (Illumina) according to the manufacturer's instructions. RNA-seq libraries for total ribosomal RNA-depleted RNA were prepared using the TruSeq Stranded Total RNA Library Prep Gold (Illumina) according to the manufacturer's instructions. cDNA libraries were sequenced using an Illumina NextSeq 500 platform.

## Viral load by quantitative real-time PCR analysis

In BSL3 experiments conducted in the BSL3 facility at the Icahn School of Medicine at Mount Sinai, Genomic viral RNA was extracted from supernatants using TRIzol reagent according to the manufacturer's instructions (Thermo Fisher). RNA was reverse transcribed into cDNA using oligo d(T) primers and SuperScript II Reverse Transcriptase (Thermo Fisher). Quantitative real-time PCR was performed on a LightCycler 480 Instrument II (Roche) using KAPA SYBR FAST qPCR Master Mix Kit (KAPA biosystems) and primers specific for the SARS-CoV-2 nsp14 transcript as described previously (**Chu et al., 2020b**; **Corman et al., 2020**). The viral load for each sample was determined using genomic viral RNA purified from viral stocks to generate a standard curve. Error bars indicate the standard error from three biological replicates.

In BSL3 experiments conducted in the BSL3 facility at the Sheba Medical Center, Total nucleic acids were extracted from all samples using MagNA Pure 96 DNA and Viral NA Small Volume Kit (Roche) according to the manufacturer protocol. Extracted RNA was transferred to 96 well PCR plate containing 20 μl of TaqPath 1-step Multiplex Master Mix No ROX (Applied Bioscience, Cat number: A28523). This was followed by a one-step RT-PCR (TaqPath COVID-19 assay kit; Thermo-Fisher).

Thereafter, the plate was sealed with MicroAmp clear adhesive strip (Applied Bioscience, Cat number: 4306311). The plate was loaded onto a QuantStudio 5 Real-Time PCR System (Applied Bioscience, Cat number: AB-A28574) and the following amplification program was used: 25 °C for 2 min, X1 cycle 53 °C for 10 min, X1 cycle 95 °C for 2 min, X1 cycle 95 °C for 3 s, followed by 60 °C for 30 s, X40 cycles Ct threshold values were presented using the following values/parameters: MS2-15,000; by cycle 37; S gene- 20,000 by cycle 37; Orf1ab- 20,000 by cycle 37; Ngene- 20,000 by cycle 37. Samples that passed the Threshold is a Ct value >37 were re-tested or considered weak positive. The viral load for each sample was determined using genomic viral RNA purified from viral stocks to generate a standard curve. Error bars indicate the standard error from three biological replicates.

## Functional annotations of gene expression

Differentially expressed genes were tested for enrichment overlap within functional gene sets. The general test for functional enrichment of the differentially expressed genes against various functional categories was done using the PANTHER tool (*Mi et al., 2019*). Enrichment p values were calculated using Fisher's exact test and corrected with familywise (Bonferroni) multiple testing correction or the Benjamini-Hochberg False discovery method as indicated.

## Drug treatments

Approximately $5×10^5$ NHBE or PPARα CRISPR-KO NHBE cells were infected with SARS-CoV-2. After 24 hr, the medium was collected and changed to bronchial epithelial growth media supplemented with 0.1% DMSO (vehicle control), 10 μM Cloperastine (Merck; C2040), 5 μM Empagliflozin (AG-CR1-3619), 1 mM Metformin (Merck; 317240), 20 μM Fenofibrate (Merck; F6020), 20 μM Rosiglitazone (Merck; R2408), 50 μM Bezafibrate (Merck; B7273), 2 μM Wy-14643 (Cayman Chemical; 70730), 50 μM Conjugated (9Z,11E)-Linoleic acid (Merck; 16413) in 50 μM Oleic Acid-Albumin (Merck; O3008), or 20 μM Fenofibrate and 4 μM Etomoxir (Cayman Chemical; 11969). Then, every 48 hr medium was collected and replenished. The medium was stored at –80 °C immediately after removal. Culture viability was assessed at the end of the experiment using Hoechst staining, compared with mock-infected cells.

## Western blot

NHBE, PPARα CRISPR-KO NHBE cells, or PPARα-OE HEK293T cells were washed in DPBS, lysed in 1 x Laemmli Loading buffer, and boiled at 100 °C; 40 μl of cleared lysate were analyzed in a pre-cast gradient polyacrylamide gel (Bolt 4 to 12%, Bis-Tris, 1.0 mm, Mini Protein Gel/ NW04120BOX, Invitrogen) using SeeBlue Plus2 Pre-stained Protein Standard (LC5925, Invitrogen) in MES SDS running buffer (B0002, Invitrogen) according to manufacturer's instructions. The proteins were transferred to a PVDF membrane (iBlot 2 Transfer Stacks, PVDF, mini/ IB24002, Invitrogen) using iBlot2 (LifeSciences). The membrane was blocked with 5% BSA (160069, MPBio) in Tris-buffered saline plus 0.1% Tween 20 (TBST) for 1 hr at room temperature. The membranes were incubated in primary antibodies overnight at 4 °C. The next day, the membranes were washed in TBST (3 × 10 min) and then incubated with horseradish peroxidase-conjugated secondary antibody for 2 hr at room temperature. After the TBST washes (4 × 10 min), EZ-ECL kit (Sartorius; 20-500-1000A, 20-500-1000B) was used to detect the HRP activity. The membrane was imaged on a Vilber Fusion FX and band densitometry was performed on FIJI.

The following commercial primary antibodies were used: anti-PPARα (1:1000;ab24509, Abcam) and anti-α-tubulin (1:2000; T6074, Sigma). Commercial horseradish peroxidase-conjugated secondary antibodies were: anti-rabbit (111-035-003, Jackson) and anti-mouse (115-035-003, Jackson). All primary antibodies were used in 5% BSA in TBST. Secondary antibodies were used at a 1:8000 dilution in TBST.

The gel, ladder, and equipment to run and transfer the gel were kindly provided by Prof. Eran Meshorer, Institute of Life Sciences, The Hebrew University of Jerusalem. The anti-tubulin and both HRP-conjugated antibodies, as well as the HRP detection kit, were kindly provided by Prof. Benjamin Aroeti, Institute of Life Sciences, The Hebrew University of Jerusalem.

## Quantification and statistical analysis

Work done in the BSL3 facility at the Icahn School of Medicine at Mount Sinai was done on NHBE from a single donor, repeated in three experimental repeats with three or more technical repeats in each experiment. Work done in the BSL3 facility at the Sheba Medical Center or in the BSL2 facility at The Hebrew University of Jerusalem was done on NHBE from two donors, repeated in three experimental repeats each (unless noted otherwise by the n value) with three or more technical repeats in each experiment. Work done in the BSL3 facility at the Sheba Medical Center in different variants was done separately and independently for each variant and repeated as listed above.

Measurements were technically repeated three or four times for each sample, images were analyzed with five or more fields of view; Graphs show mean ± SEM; Continuous variables were compared with a Mann-Whitney U test or a two-sample t-test or ANOVA. Categorical variables were compared with a chi-squared or Fisher's exact test, as appropriate. FDR correction was used to adjust for multiple comparisons and RNA seq comparisons; Hypergeometric testing was used to assess statistically significant enrichments. * indicates $p<0.05$, ** indicates $p<0.01$, *** indicates $p<0.001$, unless denoted otherwise.

## Observational studies

### Israeli study

A retrospective, multi-center study was conducted in Hadassah and Ichilov Medical Centers. A total of 150,976 participants were diagnosed positive for SARS-COV-2 following WHO interim guidance (World Health Organization, 2020). Only patients hospitalized and diagnosed with COVID-19 were included. participants with incomplete electronic medical records, aged less than 18, with pregnancy or severe medical conditions, including acute lethal organ injury (i.e. acute coronary syndrome, acute stroke, and severe acute pancreatitis) were excluded. The flowchart for patient inclusion is illustrated in *Figure 4—figure supplement 1*. Participants were admitted between March 1st, 2020, and January 31st, 2021 to either the Hadassah Medical Center in Jerusalem or the Tel Aviv Sourasky Medical Center. The final date of the follow-up was February 28th, 2021. The study protocols were approved by the institutional ethics committee. Patient informed consent was waived by each ethics committee. Demographic and clinical characteristics, vital signs, laboratory tests, medical history and comorbidities, therapeutic interventions, and outcome data were extracted from electronic medical records using a standardized data collection method. The laboratory data included routine blood tests, blood counts, and serum biochemical markers reflecting c-reactive protein, sepsis, liver injury, kidney injury, cardiac injury, glycemic status, and D-dimer were collected during hospitalization. In-hospital medication and respiratory intervention included the classification of the drugs, the dosage, the course of treatment, and using respiratory support were also extracted from medical records.

The retrospective study was designed to assess initial relationships between metabolic regulating drug use and COVID-19 clinical outcomes (28-day mortality and duration of hospitalization, ICU admission, mechanical ventilation, oxygen supplementation, disease severity at baseline, and inflammatory marker changes) versus a control group that did not take any drug of this type.

COVID-19 poses a significant risk in older patients and patients with comorbidities (*Rosenthal et al., 2020*). Hence, to account for the fact that metabolic drug users were older and had more comorbidities, we included metabolic regulating drug users and patients over 45 in the comparative analyses, creating a more comparable control group suitable for the between-treatment evaluations, as previously described (*Cummings et al., 2020*). Propensity score matching was avoided in this multi-drug comparison as it has been shown to increase model imbalance, inefficiency, model dependence, and bias in multiple group comparisons in small treatment groups. Significant differences in treatment group size and characteristics are expected to result in an underestimation of treatment effect and a high level of overt bias (*King and Nielsen, 2019*; *Wang, 2021*; *Fullerton et al., 2016*; *Ali et al., 2019*).

Comparisons were conducted between hospitalized COVID-19 patients using one or more metabolic regulating drugs (fibrates, thiazolidinediones, metformin, SGLT2 inhibitors, statins, or telmisartan [IRE1α inhibitor]) versus control patients not taking any metabolic regulating drugs. Baseline values are defined as measurements taken upon hospital admission. Statistical analyses were performed using SAS v9.4 (SAS, SAS Institute Cary, NC USA) software and R-3.6.3 (R Foundation for Statistical Computing, Vienna, Austria). Continuous variables were summarized by a median and interquartile

range (IQR) and categorical variables by a count and percentage. Statistical testing was two-sided. A p-value <0.05 was considered statistically significant. Missing data was not imputed. Nominal p-values are presented since this was an exploratory study. Demographic and baseline clinical characteristics, comorbidities, and laboratory examinations, as well as initial univariate clinical outcomes, were compared between the groups (drugs versus no drugs) by data type using a two-sample t-test or Fisher's exact test as appropriate.

The relative risk of hospitalization, ICU admission, and 28-day all-cause-mortality of COVID-19 patients versus the general hospital population (1-year period, 5-year period, and 10-year period prior to study start date in patients 30 years and older) are presented with 95% confidence interval and level of significance (Wald test).

Dynamic changes of inflammatory markers were depicted using locally weighted scatterplot smoothing (Lowess) plotting *Cleveland, 1979* from day 1 to day 21 after admission, comparing each drug group to control patients that did not take metabolic regulators.

Time-to-event data is presented with Kaplan-Meier plots. Time-to-events are measured in days from the date of hospital admission to the date of in-hospital death, and release from the hospital or last follow-up or 28 days whichever is sooner. Cox regression was performed to compare time-to-event data between the groups adjusting for covariates that may have been imbalanced between the groups. We did not perform matching since Cox regression models applied to the entire study cohort can effectively address confounding attributable to observed covariates and maximize power by using all data available. Hazard ratios are comparing drug to control group, adjusted for covariates (age, sex, current smoker, asthma, chronic obstructive pulmonary disease, cerebrovascular accident, chronic heart disease, chronic liver disease, chronic kidney disease, obesity, diabetes, hypertension, and dyslipidemia) with a level of significance and 95% confidence interval. In cases of monotone likelihood (non-convergence of likelihood function), Firth's penalized maximum likelihood bias reduction method for Cox regression was implemented. Cox Regression with Firth's Penalized Likelihood has been shown to provide a solution in the case of monotone likelihood (non-convergence of likelihood function) and was shown to outperform Wald confidence intervals in these cases (*Heinze and Schemper, 2001*).

## Italian study

A validation study was conducted by phone interviews of the last 2123 patients examined in the Outpatient Lipid Clinics of the University of Bologna and of the Niguarda Hospital in Milan during the last 12 months and on adequately dosed statins, fenofibrate, or both for at least 3 months. We excluded patients on lipid-lowering nutraceuticals (including polyunsaturated fatty acids), very low-dose or alternate-day statins, ezetimibe alone, PCSK9 inhibitors, and those on fibrates other than fenofibrate, in order to reduce the heterogeneity of the sample. Data were sampled based on comorbidities (obesity, chronic obstructive pulmonary disease, cardiovascular disease, managed as dummy variables), personal COVID history and severity, and contact with people affected by COVID. The study was carried out in adherence with the declaration of Helsinki. All participants were fully informed of the objectives of the questionnaire and gave their oral authorization to use their data for research purposes. The telephone calls were recorded. Age was compared between groups with ANOVA followed by post-hoc testing using Tukey's method. Percentages were compared by a Chi-square test followed by Fisher's exact test.

## US study

A validation study was conducted using an existing observational cohort of 920,922 veterans with hypertension (defined by diagnostic codes for hypertension and at least two fills for antihypertensive medications from January 1, 2020, to October 25, 2020, and restricted to those veterans with evidence of using the Veterans Health Administration for their primary care). There were 5144 (0.6%) veterans in the cohort who tested positive for SARS-CoV-2 between March 14, 2020, and October 25, 2020. Medication use was determined by confirmed pharmacy fills. The cohort contained a diverse, non-homogenous patient population with different disease severity. To minimize baseline differences between fenofibrate users and the three comparison groups (non-users, statin users, and TZD users), 1:5 propensity score matching was performed using Stata version 15.0. Baseline matching variables included age, sex, body mass index, race/ethnicity, and history of atherosclerotic

cardiovascular disease, heart failure, diabetes mellitus, chronic lung disease, chronic liver disease, dementia, and current or former smoker. We performed nearest neighbor matching with a caliper of 0.1. We required a<10% standardized difference in each of the matched covariates between matched groups, as well as Rubin's B of ≤25% and Rubin's R between 0.5–2 to verify sufficient matching.

## Interventional study

### Design and participants

The study was conducted as an open-label, phase 3 a clinical trial, in the Barzilai Medical Center, Ashkelon, Israel. The study was approved by the Barzilai Medical Center Research Ethics Committee (0105–20-BRZ). The study enrolled adults (≥18 years of age) with severe Covid-19 pneumonia, as confirmed by positive polymerase-chain-reaction (PCR) and evidenced by bilateral chest infiltrates on chest radiography or computed tomography. Eligible patients had a disease severity score of 4 (Hospitalized, requiring supplemental oxygen), increased oxygen requirement compared to baseline at home, a blood oxygen saturation of 93% or less on room air, or a ratio of the partial pressure of oxygen to the fraction of inspired oxygen (PaO2/FiO2) of less than 300 mm Hg, respiratory rate >30 breaths/min, and lung infiltrates >50% on chest CT within 72 hr of hospital admission or within 72 hr of a positive test result.

Individuals who had respiratory failure, septic shock, and/or multiple organ dysfunction, SOFA ≥ 5 or Disease Severity Score ≤ 3 (requiring noninvasive mechanical ventilation, requiring extracorporeal membrane oxygenation (ECMO), invasive mechanical ventilation, or all) were excluded. Additionally, individuals with known hypersensitivity to fenofibrate, patient-reported history, or electronic medical record history of severe kidney disease (defined as any history of dialysis, history of chronic kidney disease stage IV or estimated Glomerular Filtration Rate (eGFR) of <30 ml/min/1.73 m² at the time of enrollment), acute pre-renal azotemia at the time of enrollment in the opinion of the investigator or bedside clinician, most recent mean arterial blood pressure prior to enrollment <65 mmHg, patient-reported history or electronic medical record history of severe liver disease (defined as cirrhosis, history of hepatitis B or C or documented AST or ALT >10 times the upper limit of normal measured within 24 hr prior to enrollment), patient-reported history or electronic medical record history of gallbladder disease, potassium >5.0 within 24 hr prior to enrollment (unless a repeat value was ≤ 5.0), treatment with coumarin anticoagulants, immunosuppressants, or bile acid resins or female subjects breastfeeding or undergoing fertility treatments were also excluded.

All participants provided written informed consent signed by the participant or legally authorized representative. Standard care according to local practice (supplemental oxygen, antiviral treatment, anticoagulants, vitamin D3, low-dose glucocorticoids, convalescent plasma and supportive care) was provided. However, concomitant treatment with another investigational agent (except antiviral drugs) or any immunomodulatory agent, was prohibited. Written informed consent was obtained from all the patients or, if written consent could not be provided, the patient's legally authorized representative could provide oral consent with appropriate documentation by the investigator. The primary analysis was performed on day 14, a follow-up was done 28 days post-admission.

### Procedures

Participants who met the inclusion criteria were assigned to intervention with nanocrystallized fenofibrate (TriCor, AbbVie Inc, North Chicago, IL USA) at a dose of 145 mg (1 tablet) once per day. Standard care for severe-hospitalized COVID-19 patients was provided according to local practice: antiviral treatment, vitamin D3, low-dose glucocorticoids, convalescent plasma, and supportive care as well as antipyretics for symptoms of fever (products containing paracetamol, or non-steroidal anti-inflammatories such as aspirin and ibuprofen) and dextromethorphan for symptoms of cough. Standard chronic treatments were continued unless COVID-19, clinical status, or fenofibrate treatment was a contraindication for treatment. Control patients were collected from the observational study's database and filtered to patients that met the inclusion criteria, admitted with low immunoinflammatory stress (NLR <10 at admission), and treated according to the standard care used in the interventional study.

## Valuations

For the evaluation of patients in this trial, the baseline was defined as the last observation before the administration of fenofibrate on day 0. The patients' disease severity was assessed on an ordinal scale according to the following categories: The scale is as follows: (1) Death; (2) Hospitalized, on invasive mechanical ventilation or extracorporeal membrane oxygenation (ECMO); (3) Hospitalized, on non-invasive ventilation or high flow oxygen devices; (4) Hospitalized, requiring supplemental oxygen; (5) Hospitalized, not requiring supplemental oxygen; (6) Not hospitalized, limitation of activities; (7) Not hospitalized, no limitations of activities. Clinical status was recorded at baseline and every day during hospitalization.

## Viral RNA and S-gene target failure (SGTF) detection by real-time PCR

Extracted RNA was transferred to 96-well PCR plate containing 20 µl of TaqPath 1-step Multiplex Master Mix No ROX (Applied Bioscience, Cat number: A28523). This was followed by a one-step RT-PCR (TaqPath COVID-19 assay kit; Thermo-Fisher). Thereafter, the plate was sealed with MicroAmp clear adhesive strip (Applied Bioscience, Cat number: 4306311). The plate was loaded onto a Quant-Studio 5 Real-Time PCR System (Applied Bioscience, Cat number: AB-A28574) and the following amplification program was used: 25 °C for 2 min, X1 cycle 53 °C for 10 min, X1 cycle 95 °C for 2 min, X1 cycle 95 °C for 3 s, followed by 60 °C for 30 s, X40 cycles Ct threshold values were preset using the following values/parameters: MS2-15,000; by cycle 37; S gene- 20,000 by cycle 37; Orf1ab- 20,000 by cycle 37; Ngene- 20,000 by cycle 37. Samples that passed the Threshold is a Ct value >37 were re-tested or considered weak positive. Above threshold values of MS2, Orf1ab, and Ngene, but not S gene was considered S-gene target failure (SGTF). SGTF serves as a proxy for identifying B.1.1.7 cases (*Brown et al., 2021*; *Davies et al., 2021a*).

## Variant detection by real-time PCR

Allplex SARS-CoV-2 Variants I Assay from Seegene Inc was used according to the manufacturer protocol to perform rRT-PCR. Briefly, Extracted RNA (5 µl) was transferred to 96 well PCR plate containing 15 µll of the master mix. Plates were then spun down at 2500 rpm for 5 s and analyzed on a CFX96 Touch Real-Time PCR from BioRad. Reverse Transcription reaction 1 cycle: 50 °C/20 min – 95 °C/15 min. PCR reaction 45 cycles: 94 °C/15 s – 58 °C/30 sec. Gene amplifications were analyzed by FAM (E484K mutation on S-Gene), HEX (RdRP), Cal Red 610 (N501Y mutation on S-Gene), Quasar 705 (69-70del on S-Gene), and Quasar 670 (Human Endo Internal control) fluorophores. Results were compiled and analyzed using the 2019-nCoV viewer from Seegene Inc according to the manufacturer's instructions.

## Statistical analysis

Demographic data were summarized, continuous variables with non-normal distributions were expressed as median [IQR] and categorical variables were expressed as numbers and percentages (%). The sample size is detailed in each display item. Comparisons between groups were performed with Mann-Whitney U test for continuous variables and Fisher's exact test or chi-squared test for categorical variables.

Analysis of weighted differences in hospitalization duration, mortality, and incidence of oxygen weaning was done using the Mantel–Haenszel test. The cumulative rates of death and hospital discharge were compared using Kaplan-Meier curves, a log-rank test, and cause-specific Cox regression analysis. The hazard ratio (HR) was calculated using the Cox proportional hazard model comparing the treatment group versus the non-treatment group as previously described (*Cheng et al., 2020*; *Zhang et al., 2020a*). In the Cox regression models, individuals discharged were treated as '0-at risk' but not as censored data since individuals with COVID-19 would not be discharged unless their symptoms were significantly relieved and two continuous viral PCR negatives were achieved. Additionally, a clinic or electronic (medical records) follow-up at 28 days was conducted to register out-of-hospital death, need for supplemental oxygen, and/or rehospitalization. Since no deaths were recorded in the intervention group, Cox proportional hazard regression for mortality was performed using Firth's penalized maximum likelihood bias reduction method. Cox Regression with Firth's Penalized Likelihood has been shown to provide a solution in the case of monotone likelihood (non-convergence of likelihood function) and was shown to outperform Wald confidence intervals in these cases (*Heinze*

*and Schemper, 2001*). Regression adjustment was applied to remove residual confounding bias where it included the covariates with a standardized difference greater than 0.10. Multi-variable adjusted residual imbalances including age, gender, clinical characteristics on admission, indicators of disease severity and organ injuries on admission, and pre-existing medical conditions were adjusted in the analysis of the association between treatment and clinical outcomes. The proportional hazard assumptions were verified using correlation testing based on the Schoenfeld residuals.

Dynamic changes of inflammatory factors tracking from day 0 to day 8 after treatment were depicted using the Lowess model (*Cleveland, 1979*; *Shyu, 2017*). A two-side α less than 0.05 was considered statistically significant. Data were analyzed in R-3.6.3 (R Foundation for Statistical Computing, Vienna, Austria) and SPSS Statistics (version 23.0, IBM, Armonk, NY, USA). Day 0 was determined to be the first day of treatment with nanocrystallized fenofibrate in the intervention group or first-day disease severity has reached 4 (but not higher) and at least 3 MOH indicators (increased oxygen requirement compared to baseline among those on home, a blood oxygen saturation of 93% or less on room air, or a ratio of the partial pressure of oxygen to the fraction of inspired oxygen (PaO2/FiO2) of less than 300 mm Hg, respiratory rate >30 breaths/min, and lung infiltrates >50% on chest CT) were recorded.

## Ethics and oversight

All procedures performed in studies involving human participants were in accordance with the ethical standards of the institutional and/or national research committee and with the 1964 Helsinki Declaration and its later amendments or comparable ethical standards.

In the observational studies - the Israeli study was approved by the local institutional review board of the Hadassah Medical Center (IRB approval number no. HMO 0247–20) and the local institutional review board of the Ichilov Medical Center (IRB approval number no. 0282–20-TLV). The Italian study was reviewed by the local ethical board (AVEC) of the IRCSS S.Orsola-Malpighi University Hospital (approval number no. code LLD-RP2018).

The American study was reviewed by the local institutional review board of Corporal Michael J. Crescenz VA Medical Center (IRB approval number 01654).

The interventional study was conducted in accordance with the Good Clinical Practice guidelines of the International Council for Harmonisation E6 and the principles of the Declaration of Helsinki or local regulations, whichever afforded greater patient protection. The study was reviewed and approved by the Barzilai Medical Center Research Ethics Committee (0105–20-BRZ).

Statistical analysis of the Israeli studies was done by BioStats Statistical Consulting Ltd. (Maccabim, Israel), funded by the sponsor. Data management is performed in compliance with GCP and 21 CFR part 1. Statistical analyses and reporting are performed in compliance with E6 GCP, E9, and ISO 14155. Independently validated by the author. Statistical analysis of the Italian study was done by Prof. Arrigo Cicero and Dr. Chiara Pavanello. Statistical analysis of the US study was done by Prof. Jordana Cohen.

### Software resources

Our custom Cell Analysis CellProfiler Pipeline is available at https://github.com/avnere/Single-Cell-Analysis-CellProfiler-Pipeline, (copy archived at swh:1:rev:cdf361351ffbea4c43c2059a6e411d136889c1a1; *Ehrlich, 2018*).

## Results
### The metabolic fingerprint of SARS-CoV-2 infection

To elucidate the metabolic effects of SARS-CoV-2 we infected primary human bronchial epithelial cells with the virus (**methods**). Infected cells became noticeably smaller, showing vacuolization. RNA-Seq analysis of infected primary cells identified 535 differentially expressed genes (FDR <0.05). Enrichment analysis identified the regulation of viral transcription (FDR $<3 \times 10^{-2}$), immune processes (FDR $<9 \times 10^{-4}$), and cellular response to stress (FDR $<5 \times 10^{-11}$). An analysis was also carried out on RNA-Seq data obtained from primary small airway epithelial cells infected with SARS-CoV-2 (*Vanderheiden et al., 2020*), lung biopsies obtained from COVID-19 autopsies, and lung epithelial cells obtained from bronchoalveolar lavage of COVID-19 patients (*Liao et al., 2020*). All four sample groups showed similar enrichment patterns (*Figure 1A*). These four sample groups also display significant enrichment

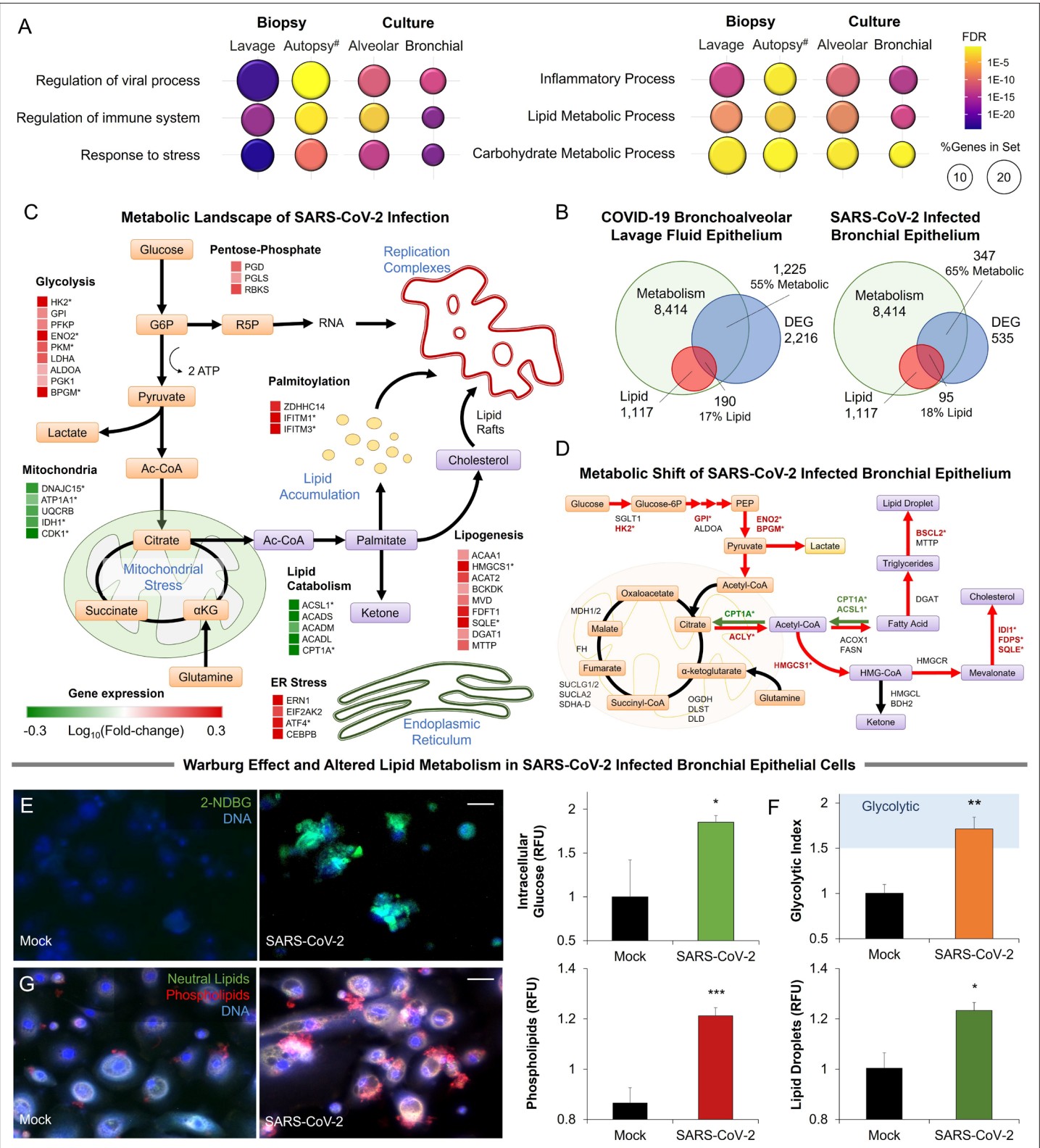

**Figure 1.** Metabolic fingerprint of SARS-CoV-2 infection. (**A**) Bubble plot visualization of GO terms enriched by SARS-CoV-2 infection. Epithelial cells were isolated by bronchoalveolar lavage from 6 severe COVID-19 patients compared to 4 healthy patients (lavage). Post-mortem lung biopsies from 2 severe COVID-19 patients compared to surgical biopsies from 2 non-COVID patients (autopsy). Culture sample groups include primary small airway epithelial cells (n=3; alveoli) and primary bronchial epithelial cells (n=3; bronchial) infected with SARS-CoV-2. Enrichment analysis shows immunoinflammatory response, cellular stress (FDR <10⁻²²), and lipid metabolism (FDR <10⁻⁵). (**B**) Venn diagram describing the relationship between

*Figure 1 continued on next page*

*Figure 1 continued*

differentially expressed genes (DEG), metabolic genes (GO:0008152), and lipid metabolism genes (GO:0006629) in SARS-CoV-2 infection of primary bronchial epithelial cells and COVID-19 patient samples. Across all four sample groups 58 ± 3% of the differentially expressed genes were metabolism-related, with 15 ± 2% of the genes associated with lipid metabolism. (C) Schematic depicting the metabolic landscape of SARS-CoV-2 infection superimposed with a heat map of pathway-associated genes. Red and green boxes indicate gene expression changes following infection in primary bronchial epithelial cells. * marks differentially regulated genes (n=3, FDR <0.05). (D) Schematic of central carbon metabolism and lipid metabolism fluxes superimposed with flux-associated genes. Differentially expressed genes (n=3, FDR <0.01) are marked with *. Genes and associated fluxes are highlighted in red or green for up- or down-regulation, respectively. (E) Microscopic evaluation of primary bronchial epithelial cells infected with SARS-CoV-2 virus or mock control shows an 85% increase in the intracellular accumulation of fluorescent glucose analog (n=3). (F) The ratio of lactate production to glucose uptake (glycolytic index) in SARS-CoV-2 and mock-infected primary cells. Index increases from 1.0 to 1.7 out of 2.0 indicating a transition to glycolysis (*i.e. Warburg effect*). (G) Microscopic evaluation of primary bronchial epithelial cells infected with SARS-CoV-2 virus or mock control. Neutral lipids (triglycerides) are dyed green while phospholipids are dyed red. Image analysis shows a 23% increase in triglycerides (n=3, p<0.05) and a 41% increase in phospholipids (n=3, p<0.001) following SARS-CoV-2 infection indicating abnormal lipid accumulation in lung epithelium. * p<0.05, ** p<0.01, *** p<0.001.# indicates a small sample size. Bar = 20 μm. Error bars indicate S.E.M.

The online version of this article includes the following source data and figure supplement(s) for figure 1:

**Source data 1.** Raw measurements, mean, standard error, and student t-test values were used to create the display items in *Figure 1*.

**Figure supplement 1.** Metabolic signature of infection in COVID-19 patients' samples and SARS-CoV-2 infected primary cells.

in metabolic processes (FDR $<4 \times 10^{-4}$), particularly lipid (FDR $<2 \times 10^{-5}$) and carbohydrate metabolic processes (FDR <0.05; *Figure 1A*).

Further transcriptional analysis shows that 58 ± 3% of differentially expressed genes are metabolism-related, with about 15 ± 2% of the genes associated with lipid metabolism (*Figure 1B*; *Figure 1—figure supplement 1*). Mapping of the SARS-CoV-2-induced transcriptional changes on the metabolic landscape of lung epithelial cells showed induction of a glycolytic phenotype (i.e*. Warburg-like effect*) and significant changes to lipid metabolism (*Figure 1C*). The shift to anaerobic metabolism is suggested to provide nucleotides for viral replication (*Mayer et al., 2019*), while changes in lipid metabolism support palmitoylation of viral proteins as well as supply lipid components of the viral replication complex (*Yan et al., 2019*; *Figure 1C*). However, in contrast to other viruses (*Levy et al., 2016*), SARS-CoV-2 infection appears to downregulate lipid catabolism (*Figure 1C*; *Figure 1—figure supplement 1*).

Mapping differentially expressed genes on the central carbon metabolism pathway showed that SARS-CoV-2 induces key glycolysis genes (*Figure 1D*) including rate-limiting enzymes such as hexoki-nase 2 (HK2) and pyruvate kinase isozyme (PKM). Interestingly, while core genes of the citric acid cycle did not change significantly, ATP citrate lyase (ACLY) was up-regulated suggesting a shift toward fatty acid synthesis. Mapping of differentially expressed genes on lipid metabolism (*Figure 1D*) showed induction of HMG-CoA synthase (HMGCS) and squalene monooxygenase (SQLE), rate-limiting steps in cholesterol synthesis (*Sharpe and Brown, 2013*). Surprisingly, we found only a few significantly up-regulated lipogenesis genes, but rather significant down-regulation of lipid catabolism genes CPT1A and ACSL1 (n=3, FDR <0.01) (*Figure 1D*).

To confirm these transcriptional signatures we validated our results in SARS-CoV-2-infected primary lung cells (*Figure 1E–G*). Microscopic analysis showed an 85% increase in intracellular glucose in infected cells (*Figure 1E*; *methods*). Concurrent metabolic analysis showed a 50% increase (n=6, <0.001) in lactate production (*Figure 1—figure supplement 1*) and a shift in the lactate over glucose ratio (glycolytic index) from 1 to 1.7 indicating a Warburg-like effect (*Figure 1F*). Alterations in lipid metabolism were confirmed by fluorescence microscopy, showing an increase of neutral lipids (n=3, p<0.05) and a significant accumulation of phospholipids (n=3, p<0.001) in SARS-CoV-2 infected primary lung cells (*Figure 1G*).

Metabolic changes are often linked to endoplasmic stress. Indeed, SARS-CoV-2 infection of primary cells induced the dsRNA-activated protein kinase R (PKR/PERK) and IRE1 pathways leading to differential expression of ATF4 and splicing of XBP1. The ATF6 pathway of ER stress was seemingly unaffected by infection. Induction of PKR/PERK and IRE1 pathways were previously shown to lead to a Warburg-like shift to anaerobic glycolysis (*Yu et al., 2014*), increased lipogenesis (*Han and Kaufman, 2016*; *Yu et al., 2013*), and decreased lipid catabolism (*Rutkowski et al., 2008*; *Figure 1—figure supplement 1*).

## SARS-CoV-2 proteins cause direct modulation of metabolic pathways

To explore the role of viral proteins in the host metabolic response to SARS-CoV-2, we expressed a large protein panel (*Gordon et al., 2020*) in primary bronchial epithelial cells (*methods; Figure 2—figure supplement 1A*). Microscopic analysis of intracellular glucose retention showed the involvement of a small subset of viral proteins including N, ORF3a, NSP7, ORF8, NSP5, and NSP12 in glucose accumulation (n=6; *Figure 2A*). Direct measurement of glucose uptake and lactate production showed a marked increase in lactate production in cells expressing the same viral protein subset (n=6, p<0.01; *Figure 2B*) confirming a viral protein-driven shift to glycolysis (n=6, p<0.01; *Figure 2C*). Independent measurement of extracellular acidification rate (ECAR), a surrogate measurement for glycolysis (*Mookerjee et al., 2017*), confirmed the activity of these viral proteins (n=6; *Figure 2D*). Mitochondrial stress test analysis (*methods*) showed a marked disruption in oxidative phosphorylation, induced by expression of N, ORF3a, and NSP7 (n=6, p<0.05; *Figure 2E–F*).

To study the role of viral proteins in lipid metabolism, we measured the exogenous fatty acid oxidation using Seahorse (*methods*) showing marked disruption in fatty acid oxidation, induced by expression of ORF9c, M, N, ORF3a, NSP7, ORF8, NSP5, and NSP12 (n=4, p<0.05; *Figure 2G*; *Figure 2—figure supplement 1B*). While triglyceride accumulation did not change, microscopic analysis confirmed a significant accumulation of phospholipids induced by expression of the same viral proteins (n=6, p<0.01; *Figure 2F*) supporting the significance of lipid accumulation for SARS-CoV-2 infection.

The inhibition of lipid catabolism by SARS-CoV-2 infection of primary lung epithelial cells and associated lipid accumulation is a unique host response (*Levy et al., 2016*) that might offer a distinct metabolic intervention. These data suggest that fibrates and other metabolic interventions that increase lipid catabolism (*Lalloyer and Staels, 2010*; *Fruchart and Duriez, 2006*) and reduce inflammatory stress (*Bocher et al., 2001*; *Sheu et al., 2002*; *Price et al., 2012*; *Ann et al., 2015*) might interfere with the virus lifecycle.

## Pharmacological modulation of SARS-CoV-2-induced metabolic pathways

The metabolic pathways induced by SARS-CoV-2 infection can be pharmacologically modulated at multiple points (*Figure 3A*; *Figure 3—figure supplement 1*). Pharmacological modulation of host metabolism was shown to block replication in other viruses (*Levy et al., 2016*; *Gualdoni et al., 2018*; *Kilbourne, 1959*; *Fujita et al., 2006*; *Ikeda et al., 2006*). SGLT inhibitors can block glucose absorption, while metformin can modulate mitochondrial activity potentially reversing a Warburg-like effect (*Andrzejewski et al., 2014*; *Tang et al., 2018*). Cholesterol synthesis can be blocked by statins, while lipid oxidation can be induced by fibrates. Telmisartan could act by decreasing ER stress through IRE1 inhibition (*Tong et al., 2016*). Thiazolidinediones are PPARγ agonists that modulate lipid content in certain tissues and are thought to reduce lung inflammation (*Ahmadian et al., 2013*; *Belvisi and Mitchell, 2009*).

Exposing primary cells infected with the alpha variant of SARS-CoV-2 to therapeutic concentrations ($C_{max}$) of these drugs produced mixed effects (*Figure 3B–E*). Rosiglitazone, empagliflozin, and metformin showed no effect at the concentrations studied. Cloperastine, a recently identified SGLT1 inhibitor (*Burggraaff et al., 2019*), reduced viral load by threefold (n=3, p0.01) without affecting cell number but did not result in a reduction of lipid content or change in the glycolytic index. However, the PPARα agonist fenofibrate blocked phospholipid accumulation (n=3, p<0.001) and the increase in glycolysis (*Figure 3B–C*). Treatment of infected primary cells with the usual therapeutic concentration of fenofibrate reduced viral load by 2-logs (n=3, p<0.001) without affecting cell number (*Figure 3D–E*).

Since the online deposition of these initial findings (*Ehrlich et al., 2020*), more recent work suggested a role for fenofibrate in blocking viral entry receptors (*Davies et al., 2021b*). To address this effect, we studied the effect of several structurally different PPARα agonists, including bezafibrate, WY14643, and conjugated linoleic acid (CLA). All four PPARα agonists showed a similar effect in both alpha and delta strains of the virus (*method*), blocking phospholipid accumulation (n=6, p<0.05; *Figure 3F*; *Figure 3—figure supplement 2*) and reducing viral load by 2–4-logs, indicating a class effect (n=6, p<0.05; *Figure 3G*; *Figure 3—figure supplement 2*).

To demonstrate the role of PPARα-induced fatty acid oxidation in our mechanism, we used etomoxir an irreversible inhibitor of CPT1A a rate-limiting enzyme in the pathway (*Figure 3A*). The addition

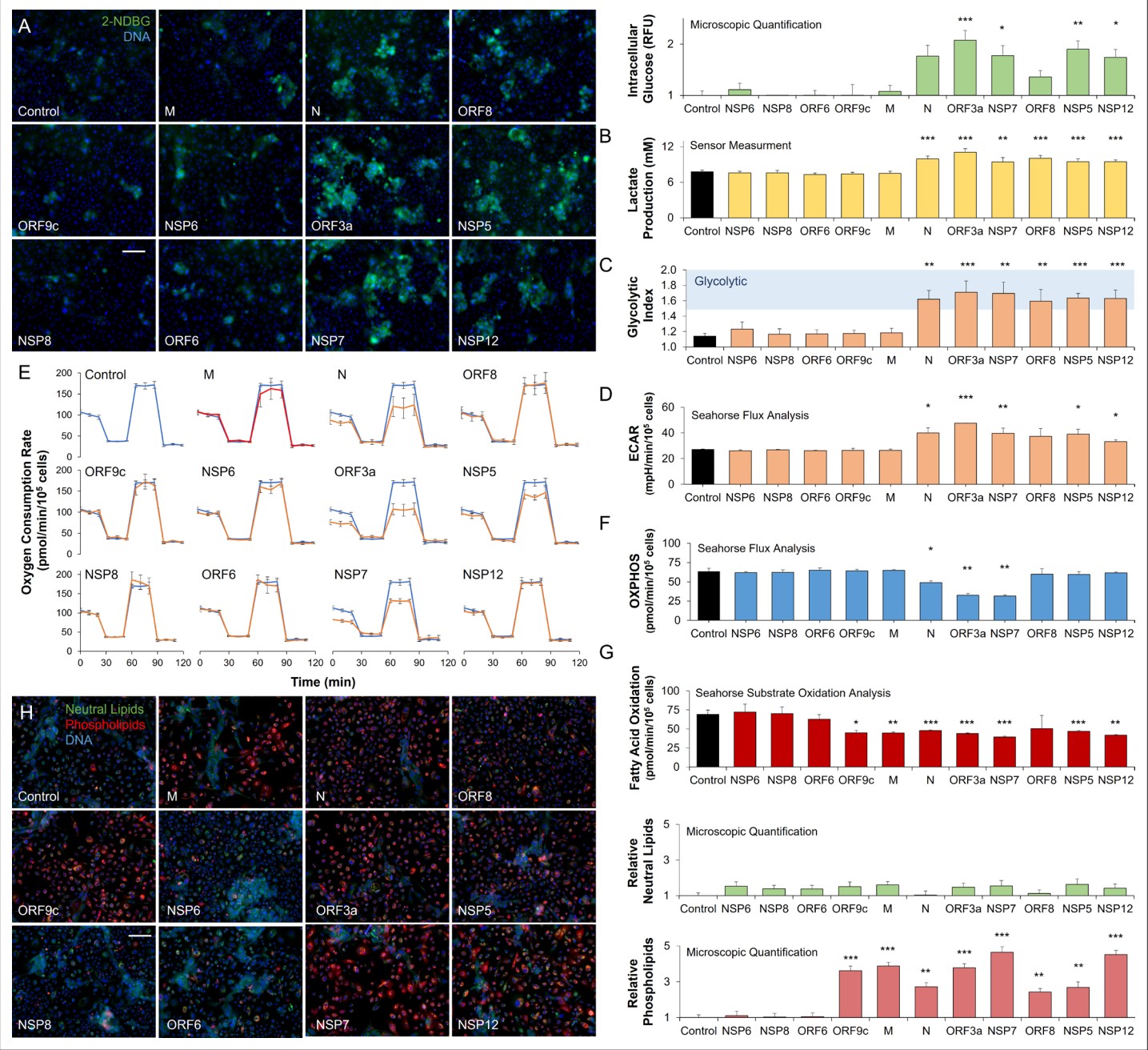

**Figure 2.** SARS-CoV-2 proteins modulate host metabolic pathways. Analysis of primary bronchial epithelial cells expressing different SARS-CoV-2 proteins for 72 hr using multiple independent assays. (**A**) Microscopic analysis shows an increased abundance of fluorescent glucose analog (2-NDBG) by a small set of viral proteins. Quantification shows a significant increase in intracellular glucose in bronchial cells expressing N, ORF3a, NSP7, ORF8, NSP5, and NSP12 (n=6, p<0.05). (**B**) Direct sensor measurement of lactate production of bronchial epithelial cells shows significantly higher lactate production (n=6, p<0.01) in cells expressing the abovementioned protein subset. (**C**) The ratio of lactate production to glucose uptake (glycolytic index) in bronchial cells expressing viral proteins. Index significantly increases from 1.1 to 1.7 marking a shift to glycolysis (n=6, p<0.01) induced by the viral proteins. (**D**) Seahorse analysis of extracellular acidification rate (ECAR) surrogate measurement for lactate production, shows independent confirmation of increased glycolysis (n=6). (**E**) Seahorse mitochondrial stress analysis of bronchial cells expressing the viral proteins. Oxygen consumption rate (OCR) is shown as a function of time. Oligomycin, FCCP, and antimycin/rotenone were injected at 25, 55, and 85 min, respectively. Orange lines indicate viral protein-expressing cells (n=6). (**F**) Quantification of oxidative phosphorylation (OXPHOS) shows a decrease of mitochondrial function following expression of N, ORF3a, and NSP7 (n=6, p<0.05). (**G**) Seahorse XF long-chain fatty acid oxidation stress analysis, a surrogate measurement for lipid catabolism, shows virus protein-induced significant decrease in lipid catabolism by ORF9c, M, N, ORF3a, NSP7, ORF8, NSP5, and NSP12 (n=4, p<0.05). (**H**) Microscopic analysis of triglycerides (neutral lipids) and phospholipids shows a virus protein-induced perinuclear lipid accumulation. Quantification

*Figure 2 continued on next page*

Figure 2 continued

shows a significant accumulation of phospholipids in cells expressing the same panel of viral proteins that induced lipid catabolism inhibition (n=6, p<0.01). * p<0.05, ** p<0.01, *** p<0.001 in a two-sided heteroscedastic student's t-test against control. Bar = 50 μm. Error bars indicate S.E.M.

The online version of this article includes the following source data and figure supplement(s) for figure 2:

**Source data 1.** Raw measurements, mean, standard error, and student t-test values were used to create the display items in *Figure 2*.

**Figure supplement 1.** Gene expression patterns of SARS-CoV-2 proteins.

of etomoxir reversed the fenofibrate effect restoring phospholipid accumulation (n=6; *Figure 3H*; *Figure 3—figure supplement 2*) and viral propagation (n=6; *Figure 3I*) in both alpha and delta strains of the virus. To further validate this pathway, we used genetic inactivation of PPARα by CRISPR KO (*methods*). Knockout of PPARα made the primary lung epithelial cells refractive to the effects of fenofibrate and etomoxir. Cells show phospholipid accumulation (n=6; *Figure 3J*; *Figure 3—figure supplement 3*) and viral propagation (n=6; *Figure 3K*) similar to untreated cells in both alpha and delta strains of the virus. Together, these data suggest that PPARα-dependent fatty acid oxidation inhibits the proliferation of SARS-CoV-2 in primary lung epithelial cells.

## Metabolic regulators affect COVID-19 severity and progression

To assess the clinical relevance of these findings we collected a total of 3233 cases of confirmed COVID-19 patients admitted to Hadassah and Ichilov Medical Centers between March 2020 to February 2021. A total of 1156 of these patients (35.8%) were registered with in-hospital use of different metabolic regulators (*Supplementary file 2*). Participants treated with metabolic regulators were older and had a higher prevalence of chronic medical conditions, including hypertension, diabetes mellitus, dyslipidemia, obesity, coronary heart disease, cerebrovascular diseases, and chronic kidney diseases than those without these treatments (*Supplementary file 2*) and thus were expected to be over-represented in ICU admissions and COVID-19-related deaths. Comparison between 2806 COVID-19 patients above the age of 30 and 532,493 recent unique hospital patient records showed a significant over-representation of patients taking thiazolidinediones, metformin, SGLT2 inhibitors, statins, or telmisartan (IRE1α inhibitor) across all COVID-19 severity indicators (*Figure 4*; *Figure 4—figure supplement 1*; *Supplementary file 2*). However, patients taking fibrates (n=21) were significantly underrepresented in hospital admissions (p=0.02) and not over-represented in other severity indicators (*Supplementary file 2*). The same trends are conserved regardless of the comparison period (*Supplementary file 2*).

Reports suggest that severe COVID-19 is characterized by early inflammation, marked by elevated C-reactive protein (CRP) *Mueller et al., 2020*, followed by distinct changes in neutrophils and lymphocytes marking the onset of the immunoinflammatory response (*Feng et al., 2020*; *Zhu et al., 2020a*). To further investigate the effect of metabolic regulators on COVID-19 progression, we tracked a subcohort of high-risk COVID-19 patients above the age of 45 that were hospitalized for 3 or more days (n=1,438; *Supplementary file 2*, *methods*). In general, fibrates use was associated with significantly shorter hospitalization duration (p=0.03; *Figure 4B*,*Supplementary file 2*). Patients taking other metabolic regulators exhibited similar or worse clinical outcomes compared to the control (*Supplementary file 2*).

To track disease progression, we followed changes in CRP during the first 21 days of hospitalization. Data were fitted using locally weighted scatterplot smoothing (Lowess) comparing each drug group to all other high-risk patients that did not take metabolic regulators (n=648; *Figure 4C–D*; *Supplementary file 2*, *methods*). Both groups had similar clinical characteristics upon admission, while comorbidities were higher in patients taking metabolic regulators (*Supplementary file 2*). High CRP levels marking systemic inflammation gradually declined after admission in the control group, reaching a plateau 14 days post-admission (*Figure 4C*). No significant differences were noted for patients taking statins, metformin, or SGLT-2 inhibitors compared with controls. CRP levels in patients taking thiazolidinediones, which is thought to increase lipid synthesis in certain tissues (*Ahmadian et al., 2013*; *Todd et al., 2007*), failed to decline. IRE1α inhibitor users exhibited slightly lower CRP levels than control patients throughout their hospitalization. Importantly, patients taking fibrates showed a significant decline in inflammation within 5 days post-admission (*Figure 4C*). The neutrophil-to-lymphocyte ratio (NLR) marking immunoinflammatory stress, rose in the control group to peak around

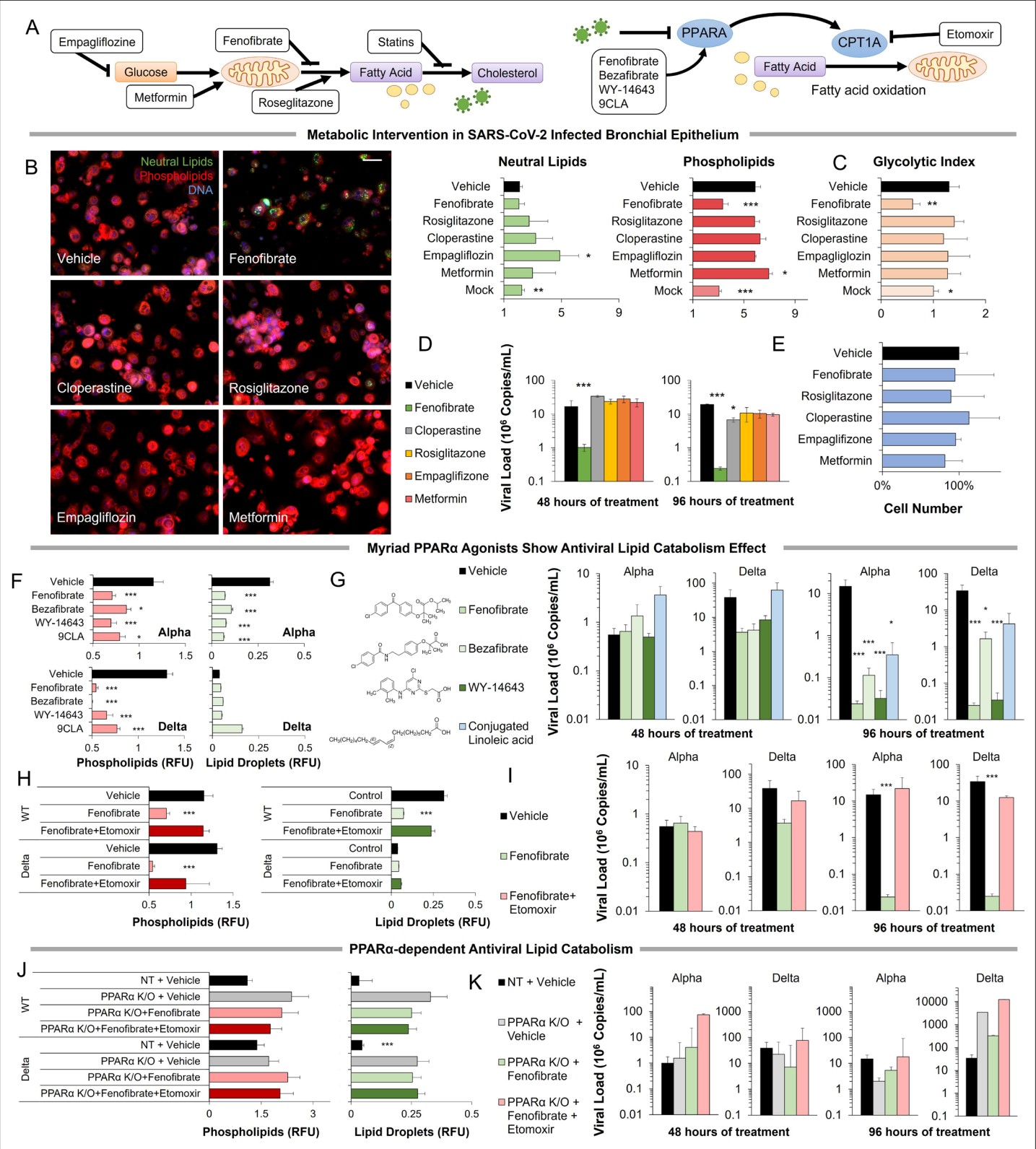

**Figure 3.** Metabolic intervention of SARS-CoV-2 shows the antiviral effect of PPARα activation. (**A**) *Left:* Schematic depicting potential drug interactions with the metabolic landscape of SARS-CoV-2 infection. *Right:* Schematic of the relationship between PPARα and fatty acid oxidation in our model. (**B**) Microscopic analysis of lipid accumulation in lung cells infected by SARS-CoV-2 (USA-WA1/2020) at MOI 2 exposed to different drugs for 96 hr compared to DMSO-treated (vehicle) and mock-infected controls. Cells treated with PPARα agonist fenofibrate showed a significant decrease in

*Figure 3 continued on next page*

*Figure 3 continued*

phospholipid content (n=3, p<0.001). (**C**) Lactate over glucose ratio of SARS-CoV-2 infected primary lung cells treated with various drugs. Fenofibrate significantly reduced the lactate-to-glucose ratio by 60% (n=3; p<0.01) normalizing the metabolic shift induced by infection. (**D**) Quantification of SARS-CoV-2 viral RNA over treatment with a physiological concentration of various drugs or DMSO (vehicle). Treatment with 20 μM fenofibrate ($C_{max}$) reduced SARS-CoV-2 viral load by 2-logs (n=3; p<0.001). Treatment with 10 μM cloperastine reduced viral load by 2.5–3-fold (n=3; p<0.05). (**E**) Cell number post-treatment was unaffected by all drugs tested. (n=3). (**F**) Microscopic analysis of lipid accumulation in lung cells infected by SARS-CoV-2 (hCoV-19/Israel/CVL-45526-NGS/2020) and B.1.617.2 variant of concern (hCoV-19/Israel/CVL-12806/2021) exposed to structurally different PPARα agonists for 5 days compared to DMSO-treated cells (vehicle). Cells treated with any PPARα agonists showed a significant decrease in phospholipid content in both viruses (n=6, p<0.001). (**G**) Quantification of SARS-CoV-2 viral RNA over treatment with a physiological concentration of various PPARα agonists or DMSO (vehicle). Treatment with 20 μM fenofibrate, 50 μM bezafibrate, or 1 μM WY-14643 reduced SARS-CoV-2 viral load by 3–5-logs (n=6; p<0.001). Treatment with 50 μM conjugated (9Z,11E)-linoleic acid and 50 μM oleic acid reduced viral load by 2.5-logs (n=6; p<0.01 in alpha variant). (**H**) Microscopic analysis of lipid accumulation in lung cells infected by SARS-CoV-2 and B.1.617.2 variant of concern (delta) exposed to PPARα agonist fenofibrate with 4 μM of lipid catabolism inhibitor, etomoxir (ETO) for 5 days compared to DMSO-treated (vehicle). Cells treated with fenofibrate showed a significant decrease in phospholipid content in both viruses (n=6, p<0.001). Phospholipid decrease was reversed by the addition of etomoxir. (**I**) Quantification of SARS-CoV-2 viral RNA exposed to the PPARα agonist fenofibrate with or without 4 μM of lipid catabolism inhibitor, etomoxir, or DMSO (vehicle). Treatment with 20 μM fenofibrate reduced SARS-CoV-2 viral load by 4–5-logs (n=6; p<0.001). Fenofibrate antiviral effect was reversed by the addition of etomoxir. (**J**) Microscopic analysis of lipid accumulation in PPARα or NT CRISPR-knockout lung cells (*methods*) infected by SARS-CoV-2 and B.1.617.2 variant of concern (delta) exposed to PPARα agonist fenofibrate with 4 μM of lipid catabolism inhibitor, etomoxir compared to DMSO-treated (vehicle). PPARα or NT CRISPR-knockout cells treated with fenofibrate did not show a decrease in phospholipid content in either virus and was unaffected by etomoxir (n=6). (**K**) Quantification of SARS-CoV-2 viral RNA after treatment with the PPARα agonist fenofibrate with or without 4 μM of lipid catabolism inhibitor, etomoxir, or DMSO (vehicle). Genetic inhibition of PPARα causes cells to be refractory to fenofibrate treatment and the addition of etomoxir (n=6). * p<0.05, ** p<0.01, *** p<0.001 in a two-sided heteroscedastic student's t-test against control. Bar = 30 μm. Error bars indicate S.E.M.

The online version of this article includes the following source data and figure supplement(s) for figure 3:

**Source data 1.** Raw measurements, mean, standard error, and student t-test values were used to create the display items in *Figure 3*.

**Figure supplement 1.** Metabolic regulators in SARS-CoV-2 infection in vitro.

**Figure supplement 2.** PPARα agonism anti-viral mechanism is ligand-wide and fatty oxidation dependent in SARS-CoV-2 infection in vitro.

**Figure supplement 3.** PPARα is required for fenofibrate rescue and etomoxir reversal in SARS-CoV-2 infection in vitro.

**Figure supplement 3—source data 1.** Complete raw and unedited blots assembly used to determine PPARα expression.

day 10 post-admission (*Figure 4D*; *Figure 4—figure supplement 2*). Treatment with SGLT2 inhibitors, metformin, or thiazolidinediones was associated with similar responses compared to controls, albeit with higher maxima for the thiazolidinedione group. Patients taking IRE1α inhibitors exhibited significantly elevated NLR post-day 10, due to decreased lymphocyte counts during recovery (*Figure 4D*; *Figure 4—figure supplement 2*). However, patients taking fibrates showed consistently low NLR throughout their hospitalization, suggesting minimal immunoinflammatory stress. Analysis of 28-day all-cause mortality showed that no deaths were reported for the small group of patients taking fibrates (n=16, *Figure 4E*). Mortality did not appear to differ for statins, IRE1α inhibitors, or metformin, but was significantly higher in patients taking SGLT2 inhibitors (aHR = 2.6; 95% CI, 1.1–6.2; p=0.034) or thiazolidinedione (aHR = 3.6; 95% CI, 1.0–12.4; p=0.043; *Figure 4E*; *Supplementary file 2*; *methods*).

Analysis of an additional observational cohort of 2123 patients examined in the Outpatient Lipid Clinics of the University of Bologna and the Niguarda Hospital in Milan during the last 12 months and on adequately dosed statins, fenofibrate, or both for at least 3 months (*Supplementary file 3*) indicates that fenofibrate users regardless of additional treatment had significantly less COVID-19 history and severe illness (*Supplementary file 3*). Additionally, in the sub-cohort of patients reporting contact with affected people, analysis indicates that statin users are significantly more likely to develop COVID-19 (p=0.02), while fenofibrate users, regardless of additional treatment are less likely to develop COVID-19 (*Supplementary file 3*). Parallel analysis of an observational cohort of 920,922 veterans with hypertension in the US Veteran's Health Administration, comparing fenofibrate to matched non-users, statins users, or thiazolidinediones users (VHA; *Supplementary file 3*), showed that fenofibrate users have shorter hospitalization duration (*Supplementary file 3*) and fairly better outcomes across several severity indicators (*Supplementary file 3*).

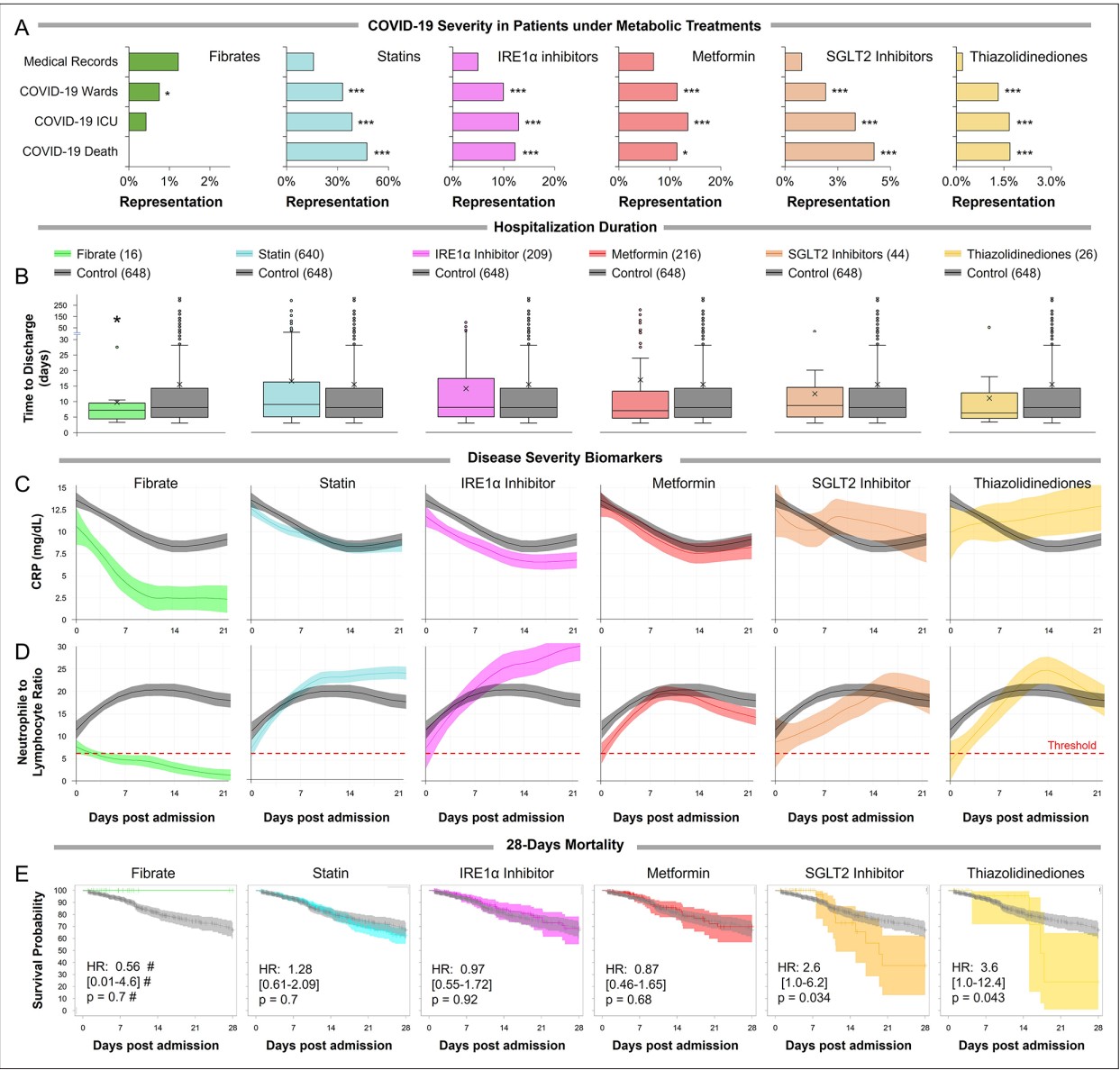

**Figure 4.** Observational study shows differential immunoinflammatory response to metabolic intervention. (**A**) Comparative representation of Israeli patients above the age of 30 taking different metabolic regulators. 532,493 unique general hospital medical records were compared with 2806 confirmed COVID-19 patients. COVID-19 patients treated with metabolic regulators were older and had a higher prevalence of chronic medical conditions and risk factors than other COVID-19 patients (***Supplementary file 2***). Patients taking thiazolidinediones (n=37; p<0.001), metformin (n=321; p<0.01), SGLT2 inhibitors (n=54; p<0.001), statins (n=924; p<0.001), or telmisartan (IRE1α inhibitor; n=278; p<0.001) were over-represented across all severity indicators (***Supplementary file 2***). Patients taking fibrates (n=21) were significantly underrepresented in hospital admissions (p=0.02) and were not over-represented in other severity indicators. * p<0.05, ** p<0.01, *** p<0.001 in a Wald test compared to the proportion of these drug users in medical records. Error bars indicate S.E.M. (**B**) Box and whisker plot of length of hospitalization in treatment and non-treatment groups (Control). Israeli patients taking bezafibrate or ciprofibrate (fibrates) were associated with significantly lower hospitalization duration (p=0.03). The numbers in parentheses indicate the number of patients. (**C–D**) Dynamic changes in the inflammation marker CRP and neutrophil-to-lymphocyte ratio (NLR) marking immunoinflammatory stress in treatment and non-treatment groups (Control) during 21-day hospitalization. The centerline shows the mean value while the 95% confidence interval is represented by the shaded region. (**C**) CRP levels gradually declined in the control group reaching a plateau by day 14 post-hospitalization. The fibrates group showed a significantly faster decline in inflammation, while the thiazolidinedione group showed marked elevation in CRP level above control. (**D**) NLR rose in the control group above normal values (dotted red line) stabilizing after 7–14 days and then declining as recovery begins. The fibrates group showed only mild stress, and maintain normal levels of NLR throughout hospitalization. Patients taking statins or IRE inhibitors showed elevated NLR post-day 10 of hospitalization. (**E**) Kaplan–Meier survival curves of 28 day in-hospital mortality for treatment and non-treatment groups (Control). The small group of patients taking fibrates did not report any deaths, while thiazolidinedione and SGLT2 inhibitor users had a significantly higher risk of mortality (HR: 3.6, 2.5; p=0.04, 0.03 respectively, ***Supplementary file 2***). * p<0.05, ** p<0.01, *** p<0.001.

*Figure 4 continued on next page*

Figure 4 continued

In boxplots, x is the mean; center line is the median; box limits are 25th and 75th percentiles; whiskers extend to 1.5×the interquartile range (IQR) from the 25th and 75th percentiles; dots are outliers. # indicates that the hazard ratios were calculated using Firth's correction for monotone likelihood with profile likelihood confidence limits.

The online version of this article includes the following figure supplement(s) for figure 4:

**Figure supplement 1.** Observational study flow diagram.

**Figure supplement 2.** The host-immune response in hospitalized COVID-19 patients in different metabolic interventions.

## Pilot study of prospective administration of nanocrystallized fenofibrate in humans with COVID-19 treated with standard-of-care

To further assess the clinical relevance of our findings, we performed an interventional single-arm clinical study in severe, hospitalized COVID-19 patients, who exhibited respiratory deterioration and severe pneumonia (NCT04661930; *methods*) (*World Health, O, 2021*). Fifteen patients (*Figure 5— figure supplement 1*; *Supplementary file 4*) were treated with 145 mg/day of nanocrystallized fenofibrate added to standard-of-care for 10 days or until discharge (*Figure 5A*) tracking multiple parameters to demonstrate differences in disease progression as observational studies by our group (*Figure 4*) and others (*Feher et al., 2021*) are not powered to show differences in endpoints such as mortality. Nanocrystallized fenofibrate was selected due to its improved lung bioavailability (*Chapman, 1987*) and short $T_{max}$ (*Ling et al., 2013*; *Maciejewski and Hilleman, 2008*) enabling rapid intervention (*Figure 5B*).

Enrolled participants exhibited a higher prevalence of chronic medical conditions compared to other hospitalized patients admitted with severe COVID-19 during the same period and treated under the same standard-of-care, who were used as historical controls (*Supplementary file 2*). Despite these comorbidities, patients treated with nanocrystallized fenofibrate exhibited a significantly shorter hospitalization (weighted difference of 2.8 days; 95% CI, 1–5.7; p<0.001; *Figure 5C*), were significantly more likely to be discharged within 28 days of hospital admission (HR = 3.6; 95% CI, 2.1–6.4; p<0.001; *Figure 5C*) and demonstrated lower rates of ICU admission and rehospitalization (*Figure 5C*; *Supplementary file 2*).

Dynamic changes in serum levels of CRP and NLR, which mark the immunoinflammatory progression of the disease, also demonstrated favorable trends (*Figure 5D–E*). Patients treated with nanocrystallized fenofibrate showed a rapid decline in CRP levels within 48 hr of treatment, with significantly lower CRP levels by day 3–5 post-treatment (p<0.001; *Figure 5D*). Immunoinflammatory stress, indicated by NLR remained muted throughout the treatment period, showing significantly lower stress by day 3–5 post-treatment (p=0.002; *Figure 5E*).

Patients treated with nanocrystallized fenofibrate also exhibited lower mortality, lower respiratory intervention rates, and significantly increased withdrawal rate from supplemental oxygen by day 7 (weighed difference of 26.1 percentage points; 95% CI, 7.0–45.2; p=0.003; *Figure 5F*; *Supplementary file 2*). COVID-19 progression was investigated as a time-varying outcome using a Cox model accounting for baseline variance, which also suggests a difference in COVID-19-related risk (*Figure 5G*; *Supplementary file 2*).

Novaplex SARS-CoV-2 variant analysis showed a dominant presence of 69/70 deletion and N501Y substitution mutation correlating to the B.1.1.7 (UK) variant of the virus in the patient population (*Figure 5H*), a similar variant distribution to the one seen in other clinical centers in Israel during the same period (*Figure 5—figure supplement 2*). Investigation into post-acute sequelae of COVID-19 in these patients, 6 months post-admission, revealed that only one patient suffered from respiratory symptoms and fatigue (IR 6.67; 95% CI, 0.1–32.0), without any additional post-acute sequelae in any of the other patients (*Figure 5I*; *Supplementary file 2*).

## Discussion

Viruses are dependent on host metabolism to obtain macromolecules essential for their lifecycle. While metabolic interventions of host pathways offer promise, the current reliance on animal models and cell lines limits our ability to identify targets for intervention due to critical metabolic and genetic differences between animal models, cell lines, and patients. In this work, we utilized primary human

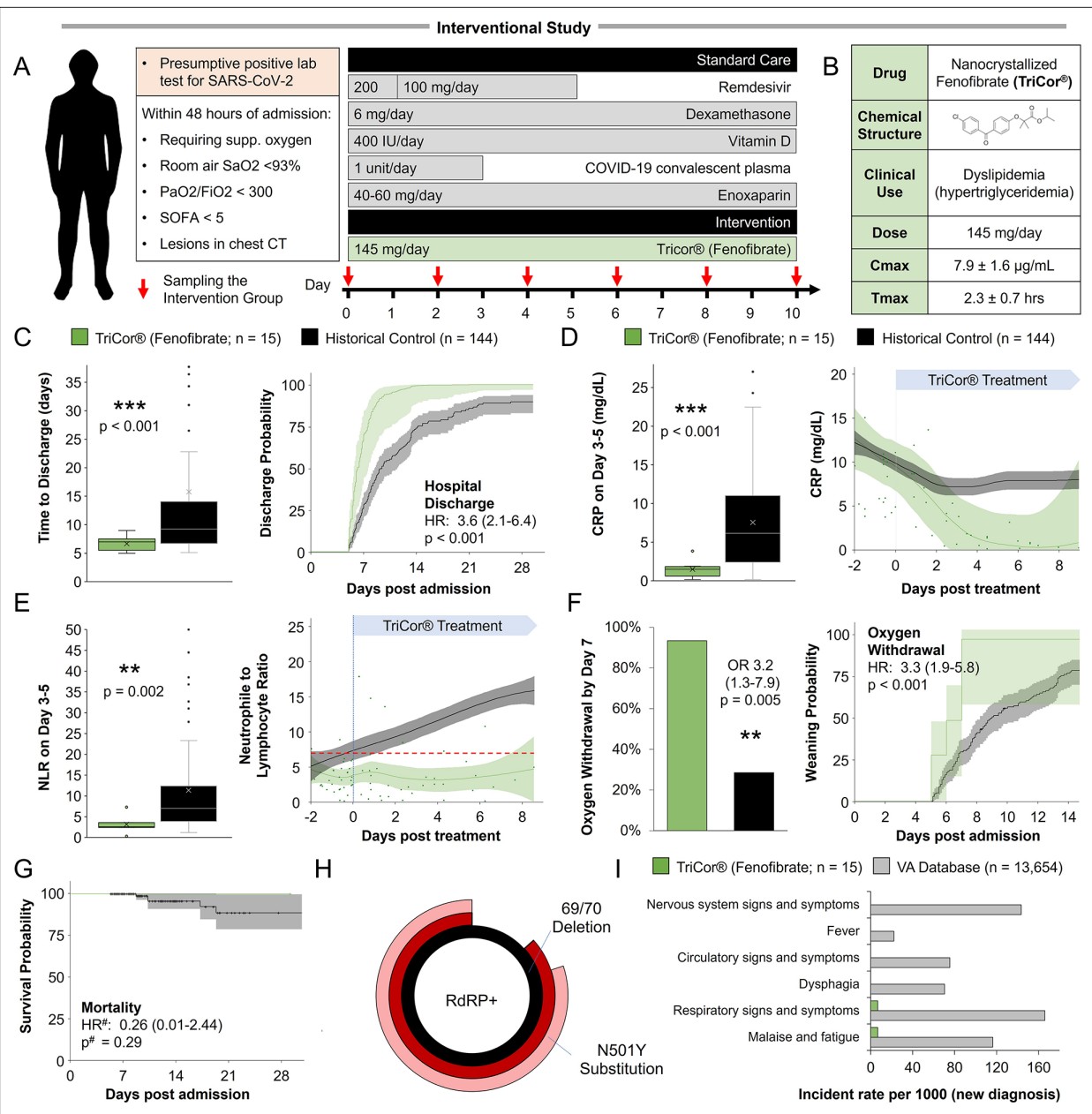

**Figure 5.** Inflammation and speed recovery in severe COVID-19 patients treated with standard-of-care plus nanocrystallized fenofibrate. (**A**) Schematic depicting interventional study design in 15 severe hospitalized COVID-19 patients receiving remdesivir, dexamethasone, and enoxaparin. Patients received 145 mg/day of nanocrystallized fenofibrate for 10 days with blood samples taken every 48–72 hr until discharge (*methods*). (**B**) Chemical, clinical, and pharmacokinetic characteristics of nanocrystallized fenofibrate. Lower $T_{max}$ compared to other fibrates enables rapid intervention in deteriorating COVID-19 patients. (**C**) Box and whisker plot of hospitalization duration (left) and Cox accumulative estimated hospital time to discharge by day 28 analysis, plotted as 1 minus the Cox estimator (right). Patients treated with nanocrystallized fenofibrate had a significantly lower hospitalization duration (n=15, p<0.001), and a higher likelihood of discharge (HR: 3.6, 95% CI 2.1–6.4, n=15, p<0.001). (**D–E**) Dynamic changes (*right*) and box and whisker plots (*left*) of immunoinflammatory indicators C-reactive protein (CRP) and neutrophil-to-lymphocyte ratio (NLR) in treatment and non-treatment groups (Control) over hospitalization duration (*methods*). The centerline shows the mean value while the 95% confidence interval is shaded. (**D**) High CRP levels gradually declined in the control group reaching a plateau by day 7. Nanocrystallized fenofibrate-treated patients showed a faster decline in inflammation, resulting in significantly lower CRP levels 3–5 days post-treatment (n=15, p<0.001). (**E**) NLR in the control group increased during hospitalization indicating severe immunoinflammatory stress. Patients treated with nanocrystallized fenofibrate showed no increase in NLR, suggesting minimal immune response, resulting in a significantly lower NLR 3–5 days post-treatment (n=15; p=0.002). (**F**) Withdrawal from oxygen support plotted as cumulative incidence at day 7 (left; OR: 3.2, 95% CI 1.3–7.9, n=15, p=0.005) Kaplan-Meier estimated time to discharge by day 28, plotted as 1 minus the survival estimator (right; HR: 2.9, 95% CI 1.7–5.0, n=15, p<0.001). (**G**) Kaplan–Meier survival curves of 28 days mortality in treatment and non-treatment groups (Control) and Cox regression modeling presenting hazard ratio estimate, 95% CI, and p-value. (**H**) Novaplex SARS-CoV-2 qPCR variant

*Figure 5 continued on next page*

*Figure 5 continued*

analysis (*methods*), showing a dominant presence of 69/70 deletion and N501Y substitution mutation correlating to the B.1.1.7 (UK) variant of the virus in the patient population. (**I**) Assessment of significant post-acute incident diagnoses in people who had been hospitalized with COVID-19 (long COVID) in patients taken from Al-Aly and colleagues (*Al-Aly et al., 2021*) compared to those treated with 145 mg fenofibrate in this study at 6 months after hospital discharge. Incident rate (IR) per 1000 at 6 months in hospitalized COVID-19 was ascertained from day 30 after hospital admission until 6 months or end of follow-up. For each outcome, cohort participants without a history of the outcome in the past year were included in the analysis. Hazard ratios (HR) and the related p-values were calculated by a Cox regression model. Odds ratios (ORs) and the related p-values were calculated using Fisher's exact test (methods). * p<0.05, ** p<.01, *** p<0.001. In boxplots, x is the mean; center line is the median; box limits are the 25th and 75th percentiles; whiskers extend to 1.5×the interquartile range (IQR) from the 25th and 75th percentiles; dots are outliers. [#] indicates that the hazard ratios were calculated using Firth's correction for monotone likelihood with profile likelihood confidence limits.

The online version of this article includes the following figure supplement(s) for figure 5:

**Figure supplement 1.** Interventional study CONSORT flow diagram.

**Figure supplement 2.** Analysis of variant emergence dynamics and distribution during the study period in participants and other hospitalized patients.

cells and clinical samples to chart SARS-CoV-2 metabolic response, to identify metabolic targets that could rapidly translate to the treatment of severe COVID-19.

Glycolysis is often upregulated to supply nucleotides for virus replication (*Levy et al., 2016*; *Gualdoni et al., 2018*; *Smallwood et al., 2017*), as part of a Warburg-like effect. We show that SARS-CoV-2 infection induced a Warburg-like effect in both bronchial and small airway primary cells, as well as COVID-19 patient samples (*Figure 1*). Recent work in Vero and Caco-2 cell lines, showed direct binding of some viral proteins to mitochondrial and glycolysis-related proteins (*Gordon et al., 2020*; *Bojkova et al., 2020*), with electron microscopy studies confirming mitochondrial disruption (*Zhu et al., 2020c*), and PET-CT studies revealing increased glycolytic activity in lungs of COVID-19 patients (*Setti et al., 2020*). These results led several groups to assess different glucose modulators as pharmacological interventions. For example, 2-Deoxyglucose (2-DG) blocked SARS-CoV-2 replication, but at concentrations 20-fold higher than $C_{max}$ while causing cellular damage (*Bojkova et al., 2020*). Our results showed a minimal effect of the SGLT1 inhibitor cloperastine and no effect of SGLT2 inhibitor empagliflozin or metformin in blocking virus replication or affecting the patient outcome (*Figures 3–4*). Recent observational studies support our findings (*Cheng et al., 2020*; *Turabian, 2020*), suggesting that while glycolysis is part of the virus lifecycle, it may not be a viable target to treat SARS-CoV-2 infection.

Our study demonstrates coordinated changes in lipid metabolism, such as the upregulation of palmitoylation and cholesterol synthesis (*Figure 1*) both critical to the virus lifecycle. SARS-CoV-2 inhibition of PPARα-dependent lipid oxidation is surprising, as the pathway was up-regulated in other viral infections (*Levy et al., 2016*). Histological analysis of COVID-19 patient biopsies confirms our findings, showing enlarged lung epithelial cells with amphophilic granular cytoplasm (*Xu et al., 2020*), while electron microscopy images of infected cells showed lipid droplet accumulation (*Zhu et al., 2020c*; *Pizzorno et al., 2020*). As lipogenesis is poorly tolerated in thin epithelial tissue, it might lead to pulmonary lipotoxicity (*Plantier et al., 2012*). Indeed, several groups looked at lipid modulators as possible pharmaceutical interventions. Triacsin C and VPS34 inhibitors blocked viral replication at concentrations 1000-fold higher than $C_{max}$ (*Silvas et al., 2020*), with similar effects shown for statins at concentrations 100-fold higher than $C_{max}$ (*Zhang et al., 2020b*). This gap might explain why observational studies on the effect of statins show an inconsistent reduction of 28 days all-cause-mortality and mixed effect regarding secondary outcomes (*Zhang et al., 2020a*). Our study confirms these earlier observations (*Figure 4*).

Fibrates are a family of amphipathic carboxylic acids that are ligands of PPARα, known to up-regulate lipid oxidation and lower serum triglycerides (*Lalloyer and Staels, 2010*; *Fruchart and Duriez, 2006*). Fibrates have also been shown to produce an anti-inflammatory and immunomodulatory effect in multiple tissues (*Bocher et al., 2001*; *Sheu et al., 2002*; *Price et al., 2012*; *Ann et al., 2015*). We show that fenofibrate inhibits viral replication in primary human lung cells, reversing phospholipid accumulation at 20 µM concentration (*Figure 3*), lower than its effective physiological concentration ($C_{max}$) recorded as 25–30 µM with a standard dose of 145 mg/day (*Wei, 2004*; *Godfrey et al., 2011*). Fenofibrate was detected at an effective plasma concentration of 15–20 µM range hours after administration (*Sonet et al., 2002*).

Our work shows that several structurally different ligands of PPARα have a similar anti-viral effect (*Figure 3*). Additionally, we show that inhibition of fatty acid oxidation reversed the antiviral effect of fenofibrate, while knockout of PPARα made the cells refractive to the drug (*Figure 3*). Recent work suggested that fenofibrate might also block viral entry receptors (*Davies et al., 2021b*). These results suggest that a combination of mechanisms might be responsible for the proposed antiviral effect of fenofibrate.

One challenge in the investigation of host metabolic pathways in vitro is the difficulty to study lipid metabolism in proliferating cell lines and stem-cell-derived models (*Levy et al., 2016*; *Alsabeeh et al., 2018*), that in addition to the *Warburg effect,* also show differences in PPARα expression and activity compared to primary cells and tissue (*Uhlén et al., 2015*; *Berglund et al., 2008*; *Karlsson et al., 2021*; *Uhlen et al., 2019*). These differences result in an effective concentration above clinical relevance, at sub-millimolar ranges (*Davies et al., 2021b*), far above the levels tested in standard high content screens (*Riva et al., 2020*; *Bakowski et al., 2021*). Thus, our work focused on primary human lung cells. In contrast to cell lines, we do not observe significant cell death in primary cell cultures even after 5 days, while the virus is clearly still replicating at this point (*Figure 3*). This is consistent with other studies in primary tissue and clinical data (*Liu et al., 2021*; *Rosa et al., 2021*; *Chu et al., 2020a*).

One challenge in the validation of our findings is that hamster models are unresponsive to fibrates (*Guo et al., 2001*; *Srivastava and He, 2010*), requiring human clinical data to support these in vitro observations. Thus, our work was directed to observational studies. We show that patients taking bezafibrate or ciprofibrate were significantly underrepresented in COVID-19-related hospitalizations (*Figure 4*). Compared to hospitalized patients that are not treated with metabolic drugs, those taking fibrates showed a minimal inflammatory response and improved disease outcomes (*Figure 4*). Other observational studies in the US and Italy showed similarly improved outcomes and disease-related complications (*Figure 4*).

The clinical importance of understanding the role of lipid metabolism in COVID-19 is further emphasized by the negative response induced by thiazolidinediones (TZD) in our study. Thiazolidinediones are ligands of PPARγ that upregulate lipogenesis in certain tissues (*Ahmadian et al., 2013*). Our study showed that Israeli COVID-19 patients taking rosiglitazone were overrepresented in ICU admissions and death (*Figure 4*), had a worse immunoinflammatory response, and had higher mortality (*Figure 4*). These results correlate with recent data showing that long-term thiazolidinedione use is associated with an increased risk of pneumonia in patients with type 2 diabetes (*Singh et al., 2011*).

To validate our findings, we carried out a prospective non-randomized interventional study of 15 severe hospitalized COVID-19 patients (NCT04661930). Severe COVID-19 patients treated with 145 mg/day of nanocrystallized fenofibrate in addition to standard-of-care showed dramatic improvement in inflammation and faster recovery compared to patients admitted during the same period in neighboring hospitals and treated with the same standard of care (*Figure 5*). This favorable course was observed despite the presence of a higher comorbidity burden in fenofibrate-treated patients. Patients treated with fenofibrate showed significantly decreased CRP levels 72 hr post-treatment suggesting a rapid decrease in inflammation, possibly due to PPARα anti-inflammatory effect. The patients NLR remained stable indicating low immunoinflammatory stress. In a 6-month follow-up, these patients report post-acute sequelae far below the rates reported in the literature (*Al-Aly et al., 2021*; *Figure 5*).

## Clinical limitations

While our clinical results are highly encouraging, baseline differences between the groups and lack of randomization must be noted. Therefore, confounding and/or random error cannot be excluded. For instance, the higher comorbidity burden in the fenofibrate group may have conditioned a lower threshold for the initial hospital admission, with consequent favorable differences in outcomes relative to controls.

In addition, it must be noted that the study controls were assigned from neighboring clinical centers serving the same diverse and mixed ethnic population, as clinical outcomes of non-consenting patients at the Barzilai Medical Center were significantly worse than the treatment group as these patients often refused or had difficulties in adhering to treatment. Thus, the best control that replicated the clinical characteristics of the patients and course of treatment, were patients that were

qualified for the study but were not included simply because they were in another local hospital. Standard of care during this time period was identical for all clinical centers in Israel.

Our work demonstrates the importance of weaving primary human cells, with clinical samples, and observational data for the rapid clinical translation of new metabolic interventions. Additional work is needed to confirm the specific activation of biochemical pathways and validate our findings in pathology samples. Still, this mechanistic understanding allowed us to design an ad hoc preliminary prospective clinical study and showed significant differences from the control group despite the small number of patients.

Our work charts the metabolic response of human lung epithelium to SARS-CoV-2 infection. Our data suggest that the up-regulation of lipid oxidation might be an effective therapeutic target in the treatment of COVID-19. A definitive answer regarding the efficacy of fenofibrate for the treatment of COVID-19 will require the execution of large randomized controlled clinical trials with meaningful clinical outcomes. Two randomized placebo-controlled trials are ongoing, including a large international trial in the US, Mexico, Greece, and several South American countries (FERMIN trial; NCT04517396), and a clinical trial in Israel (FENOC trial; NCT04661930).

## Acknowledgements

Funding was provided by European Research Council Consolidator Grants OCLD (project no. 681870) and generous gifts from the Nikoh Foundation and the Sam and Rina Frankel Foundation (YN). The interventional study was supported by Abbott (project FENOC0003). The funders had no role in study design, data collection, and interpretation, or the decision to submit the work for publication. The authors would like to thank Prof. Benjamin R tenOever and his team at the Icahn School of Medicine for providing viral load quantifications (*Figure 3D*) and fixed drug-treated SARS-CoV-2 infected primary cell cultures at their BSL3 facility at the request of the study authors.

## Additional information

### Competing interests

Avner Ehrlich: is registered as an investor in a PCT regarding the use of metabolic regulators for COVID. The author has a patent on the use of PPAR agonists to treat COVID. The author has no other competing interests to declare. Arrigo Cicero: has received personal honoraria for statistical consultation from Recipharm, and personal honoraria for manuscript writing from both Sharper Srl and Fidia Pharmaceuticals. The author has no other competing interests to declare. Cesare R Sirtori: is President of Fondazione (totally supported by family). The author has no other competing interests to declare. Jordana B Cohen: received funding from National Institutes of Health (1R01HL157108-01A1,1R01AG074989-01) . The author has no other competing interests to declare. Julio A Chirinos: has received consulting honoraria from Sanifit, Bristol Myers Squibb, Merck, Edwards Lifesciences, Bayer, JNJ, Fukuda-Denshi, NGM Bio, Mayo institute of technology and the University of Delaware, and research grants from the National Institutes of Health, Abbott, Microsoft, Fukuda-Denshi and Bristol Myers Squibb. He has received compensation from the American Heart Association and the American College of Cardiology for editorial roles, and visiting speaker honoraria from Washington University, Emory University, University of Utah, the Japanese Association for Cardiovascular Nursing and the Korean Society of Cardiology. The author is named as inventor in a University of Pennsylvania patent for the use of inorganic nitrates/nitrites for the treatment of Heart Failure and Preserved Ejection Fraction and for the use of biomarkers in heart failure with preserved ejection fraction. The author has participated on the Advisory board for Bristol-Myers Squibb Data safety monitoring board for studies by the University of Delaware and UT Southwestern, and is Vice President of North American Artery Society. The author has received research device loans from Atcor Medical, Fukuda-Denshi, Unex, Uscom, NDD Medical Technologies, Microsoft, and MicroVision Medical. The author has no other competing interests to declare. Lisa Deutsch: is affiliated with BioStats Statistical Consulting Ltd where they work as a Biostatistician. The authors has received payment for statistical work for the manuscript and consulting fees from Tissue Dynamics Ltd. The author has no other competing interests to declare. Oren Shibolet: has received consulting honoraria from Sanofi, Roche and Neopharm,

and lectures honoraria from Roche . He is the chairmen of the Israel Association for the study of the liver. The author has no other competing interests to declare. Yaakov Nahmias: is registered as an investor in a PCT regarding the use of metabolic regulators for COVID and has a patent on the use of PPAR agonists to treat COVID. The author has no other competing interests to declare. The other authors declare that no competing interests exist.

## Funding

| Funder | Grant reference number | Author |
| --- | --- | --- |
| European Research Council | 681870 | Yaakov Nahmias |
| Nikoh Foundation | | Yaakov Nahmias |
| Sam and Rina Frankel | | Yaakov Nahmias |
| Abbott | FENOC0003 | Yaakov Nahmias |

The funders had no role in study design, data collection and interpretation, or the decision to submit the work for publication.

## Author contributions

Avner Ehrlich, Conceptualization, Data curation, Formal analysis, Investigation, Methodology, Writing – original draft, Project administration, Writing – review and editing; Konstantinos Ioannidis, Data curation, Validation, Investigation, Methodology; Makram Nasar, Ismaeel Abu Alkian, Investigation; Yuval Daskal, Validation, Investigation, Visualization; Nofar Atari, Resources, Validation, Investigation; Limor Kliker, Sigal Shafran Tikva, Validation, Investigation; Nir Rainy, Resources, Validation, Investigation, Methodology; Matan Hofree, Conceptualization, Software, Validation, Investigation, Methodology; Inbal Houri, Data curation, Investigation; Arrigo Cicero, Chiara Pavanello, Resources, Data curation, Investigation; Cesare R Sirtori, Resources, Data curation, Writing – review and editing; Jordana B Cohen, Resources, Data curation, Investigation, Writing – review and editing; Julio A Chirinos, Resources, Data curation, Investigation, Methodology, Writing – review and editing; Lisa Deutsch, Software, Formal analysis, Validation, Visualization, Methodology; Merav Cohen, Formal analysis, Validation, Investigation; Amichai Gottlieb, Project administration; Adina Bar-Chaim, Michal Mandelboim, Resources, Methodology; Oren Shibolet, Conceptualization, Resources; Shlomo L Maayan, Conceptualization, Supervision, Validation, Investigation, Methodology; Yaakov Nahmias, Conceptualization, Data curation, Formal analysis, Supervision, Funding acquisition, Validation, Investigation, Methodology, Writing – original draft, Project administration, Writing – review and editing

## Author ORCIDs

Avner Ehrlich ⓘ http://orcid.org/0000-0002-6339-1665
Konstantinos Ioannidis ⓘ http://orcid.org/0000-0002-3998-1171
Yuval Daskal ⓘ http://orcid.org/0000-0003-0245-6733
Chiara Pavanello ⓘ http://orcid.org/0000-0001-5892-9857
Lisa Deutsch ⓘ http://orcid.org/0000-0002-2445-7561
Merav Cohen ⓘ http://orcid.org/0000-0001-7802-0499
Yaakov Nahmias ⓘ http://orcid.org/0000-0002-6051-616X

## Ethics

Clinical trial registration NCT04661930.
Human subjects: All procedures performed in studies involving human participants were in accordance with the ethical standards of the institutional and/or national research committee and with the 1964 Helsinki Declaration and its later amendments or comparable ethical standards. In the observational studies - the Israeli study was approved by the local institutional review board of the Hadassah Medical Center (IRB approval number no. HMO 0247-20) and the local institutional review board of the Ichilov Medical Center (IRB approval number no. 0282-20-TLV). The Italian study was reviewed by the local ethical board (AVEC) of the IRCSS S.Orsola-Malpighi University Hospital (approval number no. code LLD-RP2018). The American study was reviewed by the local institutional review board of the Corporal Michael J. Crescenz VA Medical Center (IRB approval number 01654). The interventional study was conducted in accordance with the Good Clinical Practice guidelines of the International Council for Harmonisation E6 and the principles of the Declaration of Helsinki or local regulations,

whichever afforded greater patient protection. The study was reviewed and approved by the Barzilai Medical Center Research Ethics Committee (0105-20-BRZ). Statistical analysis of the Israeli studies was done by BioStats Statistical Consulting Ltd. (Maccabim, Israel), funded by the sponsor. Data management is performed in compliance with GCP and 21 CFR part 1. Statistical analyses and reporting are performed in compliance with E6 GCP, E9, and ISO 14155. Independently validated by the author. Statistical analysis of the Italian study was done by Prof. Arrigo Cicero and Dr. Chiara Pavanello. Statistical analysis of the US study was done by Prof. Jordana Cohen.

## Decision letter and Author response

Decision letter https://doi.org/10.7554/eLife.79946.sa1
Author response https://doi.org/10.7554/eLife.79946.sa2

---

# Additional files

## Supplementary files

• Supplementary file 1. Differentially expressed genes (DEG) analysis in SARS-CoV-2 infected human lung epithelium. (Tab 2) Normal bronchial epithelial cells (Tab 3) Lung Biopsies (Tab 4) Small airway (Tab 5-6) Epithelial cells in bronchial alveolar lavage fluid. (Tab 7) Primer list used for qPCR gene expression validations.

• Supplementary file 2. Observational study descriptive statistics. (Tab 1-8) Characteristics of COVID-19 patients in the cohort. SBP, systolic blood pressure; DBP, diastolic blood pressure; COPD, chronic obstructive pulmonary disease; SpO2, oxygen saturation; ECMO, extracorporeal membrane oxygenation; IQR, interquartile range. Continuous variables were compared with a two-sample t-test and categorical variables with Fisher's exact test. (Tab 9-11) Observational comparison between unique patients' visits to Hadassah Medical Center taking metabolic regulators and unique patients in various hospitalization conditions in patients with COVID-19 taking metabolic regulators in different periods. Patients taking thiazolidinediones (n=37; $P<0.001$), metformin (n=321; $P<0.01$), SGLT2 inhibitors (n=54; $P<0.001$), statins (n=924; $P<0.001$), or telmisartan (IRE1α inhibitor; n=278; $P<0.001$) were over-represented across all severity indicators, while patients taking fibrates (n=21) were significantly underrepresented in hospital admissions ($P=0.02$) and were not over-represented in other severity indicators regardless of the period examined. Observational comparison between unique patients visiting Hadassah Medical Center during (Tab 9) 11/2018–2019, (Tab 10) 11/2015–2020, or (Tab 11) 11/2010–2020 taking metabolic regulators and unique patients in various hospitalization conditions in patients with COVID-19 taking metabolic regulators. (Tab 12-20) Characteristics of patients included in the study. Patients included in the study were between the age of 45–100, that were hospitalized for more than 3 days (N=1,438). SBP, systolic blood pressure; DBP, diastolic blood pressure; COPD, chronic obstructive pulmonary disease; SpO2, oxygen saturation; ECMO, extracorporeal membrane oxygenation; IQR, interquartile range Continuous variables were compared with a two-sample t-test and categorical variables with Fisher's exact test. (Tab 21) Cox regression model of 28days mortality in the treatment groups versus control. Adjusted HR and p-values were calculated based using a Cox regression model adjusting for age, gender, and pre-existing comorbidities (smoking, asthma, COPD, DM, hypertension, diabetes, coronary heart disease, obesity, dyslipidemia, cerebrovascular disease, chronic liver disease, and chronic kidney disease). There were no deaths recorded in fibrate patients, resulting in monotone likelihood (non-convergence of likelihood function, Firth's penalized maximum likelihood bias reduction method was implemented to calculate hazard ratios and confidence intervals). Thiazolidinediones and SGLT2 inhibitors users show a significantly higher risk of death within 28 days of hospitalization (adjusted risk). # indicates that the hazard ratios were calculated using Firth's correction for monotone likelihood with profile likelihood confidence limits.

• Supplementary file 3. International comparative validation cohorts descriptive statistics. (Tab 1) Comparative Cohort of the Outpatient Lipid Clinics of the University of Bologna and of the Niguarda Hospital in Milan. (A) Characteristics of included patients stratified by lipid-lowering treatment. A cohort of 2,123 patients (M: 48.1%, F: 51.9%) on statins (1,791, mean age 59.2±15.2 years), fenofibrate (220, mean age 60.7±15.4 years) or both (112, mean age 62.9±16.3 years) were interviewed. Patients on statins were significantly younger than those on both drugs ($P=0.023$). 177 patients received a diagnosis of COVID by molecular swab: 9.2% of statin-treated subjects, 3.2% of fenofibrate-treated ones, and 5.4% of those treated with both statins and fenofibrate ($P=0.005$) without differences in the source of exposition (family members, co-workers; $P=0.648$). (B) Disease severity stratified by lipid-lowering treatment. 134 reported mild COVID-19 symptoms and

31 patients reported severe COVID-19 symptoms, requiring hospitalization: 1.7% of statin-treated subjects, 0.5% of fenofibrate-treated ones, and 0% of those treated with both statins and fenofibrate ($P$=0.022) without differences in the source of exposure. (C) Characteristics of included patients according to the personal history of COVID. Patients affected by COVID were more frequently obese, with COPD and/or cardiovascular diseases, and had strict contact with COVID-affected subjects, independent of the lipid-lowering treatment. (D) Characteristics of patients exposed to contact with affected people (n=254). Out of 254 patients reporting contact with affected people, 45 became positive for COVID. 93.3% were in treatment with statins 4.4% with fenofibrate and 4.5% with both ($P$=0.059). Affected to exposed to positive contacts ratio according to background lipid-lowering was 20.5% in patients treated with statins 7.4% with fenofibrate and 4.5% with both ($P$=0.059). (Tab 2) Comparative cohort in the American veteran's health administration (VHA) registry. (A) Characteristics of fenofibrate users compared with non-users before and after PSM. (B) The median duration of hospitalization among fenofibrate users vs. non-users. (C) SARS-CoV-2 infection and COVID-19 severity among fenofibrate users vs. non-users.

• Supplementary file 4. Interventional study descriptive statistics. (Tab 1) Characteristics of patients compared in the patients in the interventional study 15 Participants who met the inclusion criteria were assigned to intervention with nanocrystallized fenofibrate (TriCor, AbbVie Inc, North Chicago, IL USA) at a dose of 145 mg (1 tablet) once per day. Standard care for severe-hospitalize COVID-19 patients was provided according to local practice: antiviral treatment, vitamin D3, low-dose glucocorticoids, convalescent plasma, and supportive care as well as antipyretic for symptoms of fever (products containing paracetamol, or non-steroidal anti-inflammatories such as aspirin and ibuprofen) and dextromethorphan for symptoms of cough. Standard chronic treatments were continued unless COVID-19, clinical status, or fenofibrate treatment was a counterindication for the treatment. Control patients were collected from the observational study's database and filtered to patients that meet the inclusion criteria, were admitted with low immunoinflammatory stress (NLR <10 at admission), and were treated according to the standard care used in the interventional study. SBP, systolic blood pressure; DBP, diastolic blood pressure; COPD, chronic obstructive pulmonary disease; SpO2, oxygen saturation; ECMO, extracorporeal membrane oxygenation; IQR, interquartile range. Continuous variables were expressed as median [IQR] and were compared with a Mann-Whitney U test. Categorical variables were expressed as a count and percentage (%) and compared with a chi-squared test or Fisher's exact test. The sample size is detailed in each display item. (Tab 2) Cox regression model of 28-days mortality in the treatment group versus control. Adjusted HR and p-values were calculated based using an unadjusted Cox regression model, a Cox regression model adjusting for age, gender, and pre-existing comorbidities (smoking, asthma, COPD, DM, hypertension, diabetes, coronary heart disease, obesity, dyslipidemia, cerebrovascular disease, chronic liver disease, and chronic kidney disease) or a Cox regression model adjusting for significantly different patient characteristics, obesity, chronic kidney disease, and SpO2. (A) Cox regression model of 28 days hospital discharge. (B) Cox regression model of 28 days oxygen withdrawal. (C) Cox regression model of 28 days mortality. There were no deaths recorded in the treatment patients, resulting in monotone likelihood (non-convergence of likelihood function, Firth's penalized maximum likelihood bias reduction method was implemented to calculate hazard ratios and confidence intervals).

• MDAR checklist

• Reporting standard 1. Strengthening the reporting of observational studies in epidemiology (STROBE) reporting standards used in the observetional studies.

• Reporting standard 2. Consolidated Standards of Reporting Trials (CONSORT) reporting standards used in the interventional studies.

## Data availability

Software resources: Our custom Cell Analysis CellProfiler Pipeline is available on https://github.com/avnere/Single-Cell-Analysis-CellProfiler-Pipeline, (copy archived at swh:1:rev:cdf361351ffbea4c43c2059a6e411d136889c1a1).

The following previously published datasets were used:

| Author(s) | Year | Dataset title | Dataset URL | Database and Identifier |
|---|---|---|---|---|
| tenOever BR, Blanco-Melo D | 2020 | Transcriptional response to SARS-CoV-2 infection | https://www.ncbi.nlm.nih.gov/geo/query/acc.cgi?acc=GSE147507 | NCBI Gene Expression Omnibus, GSE147507 |
| Liao M, Liu Y, Yuan J, Wen Y, Xu G, Zhao J, Cheng L, Li J, Wang X, Wang F, Liu L, Amit I, Zhang S, Zhang Z | 2020 | Single-cell landscape of bronchoalveolar immune cells in COVID-19 patients | https://www.ncbi.nlm.nih.gov/geo/query/acc.cgi?acc=GSE145926 | NCBI Gene Expression Omnibus, GSE145926 |
| Vanderheiden A, Ralfs P, Chirkova T, Upadhyay A, Zimmerman M, Danzy S, Pellegrini K, Manfredi C, Sorscher E, Mainou B, Lobby J, Kohlmeier J, Lowen A, Shi P, Menachery V, Anderson L, Grakoui A, Bosinger S, Suthar MS | 2020 | Primary Human Airway Epithelial Cultures infected with SARS-CoV-2 | https://www.ncbi.nlm.nih.gov/geo/query/acc.cgi?acc=GSE153970 | NCBI Gene Expression Omnibus, GSE153970 |

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
