## [Editor Report]

In this study, a metabolism-related drug screen showed that fenofibrate reversed lipid accumulation and blocked SARS-CoV-2 replication through a PPARα-dependent mechanism in both α and δ variants. Patients taking fibrates displayed significantly lower markers of inflammation and experienced faster recovery from disease. The data offer significant support for the concept that PPARα should be considered as a potential therapeutic approach for SARS-CoV-2 infection and emphasizes the need to complete studies of fenofibrate in large randomized controlled clinical trials.

---

## [Decision Letter]

**Decision letter after peer review:**

Thank you for submitting your article "Efficacy and safety of metabolic interventions for the treatment of severe COVID-19: in vitro, Observational, and Non-Randomized Open Label Interventional Study" for consideration by *eLife*. Your article has been reviewed by 2 peer reviewers, and the evaluation has been overseen by a Reviewing Editor and David James as the Senior Editor. The following individual involved in review of your submission has agreed to reveal their identity: Jeremy Luban (Reviewer #1).

Essential revisions:

1) Please articulate if there was a "reagent control" for the PPARalpha KO experiments, and provide such control data if available.

2) Please provide information available relating to mechanisms that are at play in how the virus is mediating the metabolic effects or the proteins involved in this process.

*Reviewer #1 (Recommendations for the authors):*

The authors performed RNA-Seq on primary bronchial and small airway epithelial cells that had been challenged in the lab with SARS-CoV-2, as well as lung biopsies and bronchoalveolar lavage from COVID-19 patients. All four groups showed enrichment for genes in lipid and carbohydrate metabolism, with increases in expression of genes encoding rate-limiting enzymes in glycolysis and the synthesis of fatty acids and cholesterol, increase in ER stress, and decreased expression of lipid catabolism genes. Based on these findings the authors hypothesized that interventions which decrease ER stress or increase lipid catabolism would inhibit SARS-CoV-2 replication. The authors then examined the effect of five drugs that block lipid metabolism and found that fenofibrate significantly decreased viral load without obvious toxicity. Since fenofibrate is a PPARa agonist, and it has been reported to block SARS-CoV-2 via direct effects on Spike and ACE2, other PPARa agonists were tested and also shown to inhibit SARS-CoV-2. The mechanistic importance of PPARa was supported by the finding that PPARa knockout, and Etoxomir, a drug which blocks a downstream target of PPARa, both blocked the antiviral effect of fenofibrate.

To follow up their lab observations, hospital records for people with COVID-19 were examined for association of clinical outcomes with the use of fibrates and other drugs including metformin, SGLT2 inhibitors, thiazolidinediones, statins, and IRE1a inhibitor. COVID-19 patients taking fibrates were underrepresented among hospitalizations, ICU admissions, and deaths. Among people hospitalized with COVID-10, CRP and neutrophil:lymphocyte ratio normalized faster and survival probability was far better with fibrates than with any other of the drugs. This study was done in Israel and complementary observational data was obtained at sites in Italy and in the US.

Finally, in a small clinical trial, fenofibrate was given to 15 patients admitted to hospital with severe COVID-19 and outcomes were compared to historical controls. Patients treated with fenofibrate had shorter hospitalization, were more likely to be discharged within 28 days, had lower rate of ICU admission, and CRP and neutrophil:lymphocyte ratio were significantly improved. While this was a small trial, and it was not a randomized trial, these preliminary results are exciting and provide optimism that the larger clinical trials currently in place will confirm the findings here.

This very interesting paper presents a well-designed set of experiments that convincingly show the importance of PPARalpha for SARS-CoV-2 replication and pathogenesis, and for the utility of currently approved PPARalpha agonists in the treatment of severe COVID-19.

*Reviewer #2 (Recommendations for the authors):*

In this work, the authors use multiple tools to evaluate their hypothesis, which includes in vitro studies of primary lung bronchiole and small airway epithelial cells, observational studies of more than 3,000 patients, comparative epidemiological analysis from cohorts in Italy and the Veterans Health Administration in the United States, and prospective non-randomized interventional open-label study.

The work follows the hypothesis that the metabolic pathway has a significant role in the SARS-CoV-2 viral infection. The authors did extensive work to validate their hypothesis, which resulted in a clinical trial, which seems to be successful and reduces significantly the severity of SARS-CoV-2 infection.

This is a very important study, as it is: 1. Tackle a significant clinical issue (Covid-19) and offer a potential treatment to reduce its severity. 2. Demonstrate an example of a scientific process that starts as an in vitro study, goes throw an observational one, and ends in a clinical trial. 3. Offers a potential mechanism of action for the SARS-CoV-2.

Overall, I think that this is a very strong study with significant relevance, and I would strongly recommend accepting it.

There are some points that I think could strengthen the work (although it is very extensive):

1. As the paper focus on the metabolic effect of SARS-CoV-2, it could be a nice addition if the authors could pinpoint how the virus (or which set of proteins) is modulating the metabolic effect.

2. In the observational data (Figure 3) the authors show that thiazolidinedione (TZD) produces a negative effect. It would be nice to elaborate on this point, as the action mechanism if this drug is related to the activation of PPARgama, which is related to similar metabolic pathways that were mentioned in this study.

---

## [Author Response]

Reviewer #1 (Recommendations for the authors):The authors performed RNA-Seq on primary bronchial and small airway epithelial cells that had been challenged in the lab with SARS-CoV-2, as well as lung biopsies and bronchoalveolar lavage from COVID-19 patients. All four groups showed enrichment for genes in lipid and carbohydrate metabolism, with increases in expression of genes encoding rate-limiting enzymes in glycolysis and the synthesis of fatty acids and cholesterol, increase in ER stress, and decreased expression of lipid catabolism genes. Based on these findings the authors hypothesized that interventions which decrease ER stress or increase lipid catabolism would inhibit SARS-CoV-2 replication. The authors then examined the effect of five drugs that block lipid metabolism and found that fenofibrate significantly decreased viral load without obvious toxicity. Since fenofibrate is a PPARa agonist, and it has been reported to block SARS-CoV-2 via direct effects on Spike and ACE2, other PPARa agonists were tested and also shown to inhibit SARS-CoV-2. The mechanistic importance of PPARa was supported by the finding that PPARa knockout, and Etoxomir, a drug which blocks a downstream target of PPARa, both blocked the antiviral effect of fenofibrate.To follow up their lab observations, hospital records for people with COVID-19 were examined for association of clinical outcomes with the use of fibrates and other drugs including metformin, SGLT2 inhibitors, thiazolidinediones, statins, and IRE1a inhibitor. COVID-19 patients taking fibrates were underrepresented among hospitalizations, ICU admissions, and deaths. Among people hospitalized with COVID-10, CRP and neutrophil:lymphocyte ratio normalized faster and survival probability was far better with fibrates than with any other of the drugs. This study was done in Israel and complementary observational data was obtained at sites in Italy and in the US.Finally, in a small clinical trial, fenofibrate was given to 15 patients admitted to hospital with severe COVID-19 and outcomes were compared to historical controls. Patients treated with fenofibrate had shorter hospitalization, were more likely to be discharged within 28 days, had lower rate of ICU admission, and CRP and neutrophil:lymphocyte ratio were significantly improved. While this was a small trial, and it was not a randomized trial, these preliminary results are exciting and provide optimism that the larger clinical trials currently in place will confirm the findings here.

Thank you. The reviewer’s remarks are much appreciated.

Reviewer #2 (Recommendations for the authors):In this work, the authors use multiple tools to evaluate their hypothesis, which includes in vitro studies of primary lung bronchiole and small airway epithelial cells, observational studies of more than 3,000 patients, comparative epidemiological analysis from cohorts in Italy and the Veterans Health Administration in the United States, and prospective non-randomized interventional open-label study.The work follows the hypothesis that the metabolic pathway has a significant role in the SARS-CoV-2 viral infection. The authors did extensive work to validate their hypothesis, which resulted in a clinical trial, which seems to be successful and reduces significantly the severity of SARS-CoV-2 infection.

Thank you. The reviewer’s remarks are much appreciated.

This is a very important study, as it is: 1. Tackle a significant clinical issue (Covid-19) and offer a potential treatment to reduce its severity. 2. Demonstrate an example of a scientific process that starts as an in vitro study, goes throw an observational one, and ends in a clinical trial. 3. Offers a potential mechanism of action for the SARS-CoV-2.Overall, I think that this is a very strong study with significant relevance, and I would strongly recommend accepting it.There are some points that I think could strengthen the work (although it is very extensive):1. As the paper focus on the metabolic effect of SARS-CoV-2, it could be a nice addition if the authors could pinpoint how the virus (or which set of proteins) is modulating the metabolic effect.

We thank the reviewer for their comments. We now include a wide metabolic analysis displaying the metabolic outcome of different viral proteins expression in primary cells. We show that a subset of viral proteins cause lipid accumulation (Figure 2H), inhibits PPARα activity (Supp. Figure S2) and lipid oxidation (Figure 2G) and upregulates SARS-CoV-2 related immunoinflammatory markers (Figure 2I, Supp. Figure S2).

2. In the observational data (Figure 3) the authors show that thiazolidinedione (TZD) produces a negative effect. It would be nice to elaborate on this point, as the action mechanism if this drug is related to the activation of PPARgama, which is related to similar metabolic pathways that were mentioned in this study.

We thank the reviewer for their comments. We show lipid accumulation induced by PPARγ agonist rosiglitazone, increases immunoinflammatory markers in primary lung epithelial cells (Supp. Figure S7).